# Neurocomputational mechanisms of confidence in self and others

Dan Bang [1,2✉], Rani Moran [1,3], Nathaniel D. Daw [4] & Stephen M. Fleming [1,3,5✉]

Computing confidence in one's own and others' decisions is critical for social success. While there has been substantial progress in our understanding of confidence estimates about oneself, little is known about how people form confidence estimates about others. Here, we address this question by asking participants undergoing fMRI to place bets on perceptual decisions made by themselves or one of three other players of varying ability. We show that participants compute confidence in another player's decisions by combining distinct estimates of player ability and decision difficulty – allowing them to predict that a good player may get a difficult decision wrong and that a bad player may get an easy decision right. We find that this computation is associated with an interaction between brain systems implicated in decision-making (LIP) and theory of mind (TPJ and dmPFC). These results reveal an interplay between self- and other-related processes during a social confidence computation.

[1] Wellcome Centre for Human Neuroimaging, University College London, London WC1N 3BG, UK. [2] Department of Experimental Psychology, University of Oxford, Oxford OX1 3UD, UK. [3] Max Planck UCL Centre for Computational Psychiatry and Ageing Research, University College London, London WC1B 5EH, UK. [4] Princeton Neuroscience Institute and Department of Psychology, Princeton University, Princeton, NJ 08544, USA. [5] Department of Experimental Psychology, University College London, London WC1H 0AP, UK. ✉email: danbang.db@gmail.com; stephen.fleming@ucl.ac.uk

A key feature of adaptive behaviour is an ability to estimate confidence in our decisions in the absence of immediate feedback. For example, we may recognize that a choice was based on weak evidence and change our mind as conflicting but stronger evidence comes to light[1,2]. Recent years have seen substantial progress in our understanding of decision confidence. At a computational level, an emerging consensus is that our confidence in a choice reflects the probability that the choice is correct as computed using Bayesian inference[3–7]. At a neural level, converging evidence points to a central role for the pre-frontal cortex in the computation of decision confidence[8–10], the mapping of this internal estimate onto explicit reports[11,12], and the integration of single-trial confidence estimates into a long-run estimate of task performance[13].

Humans are, however, highly social and many situations require that we compute confidence not only in our own deci-sions but also in those of others. In competitive tasks, having a sense of who is more likely to succeed is useful for deciding whether to compete or opt out[14]. In cooperative tasks, having a sense of who is more likely to be correct is critical for assigning appropriate weight to others' advice[15,16], or deciding whose opinion to follow[17]. In addition, the confidence that others report is often biased[3], and can serve to manage social influence[12,18–20], and it is therefore useful to independently verify such reports. However, in contrast to the case of confidence in one's own decisions, little is known about the neurocomputational basis of confidence in decisions made by others.

Intuitively, the starting point for a social confidence compu-tation is the same as in the individual case[4]. We should first assess the difficulty of the decision: the easier the decision is, the higher the probability that others will get it right. However, it is typically not sufficient to simply simulate which choice we would have made and how confident we would have felt about this choice. Instead, we should recognise that others' ability may be different from our own: what is hard for us may be easy for others and what is easy for us may be hard for others[4]. Previous studies have investigated how we track others' ability[15,21,22], but it remains unknown whether, or how, we integrate knowledge about others' ability with our own assessment of decision difficulty when computing confidence in others' choices. Critically, it is only by taking both of these factors into account that we can predict that a high-ability individual may get a difficult decision wrong and that a low-ability individual may get an easy decision right.

This requirement for integrating estimates of both decision difficulty and others' ability predicts that multiple brain systems support confidence in others' decisions. The first component, the estimation of decision difficulty, is computationally similar in both the individual and the social case and may therefore be supported by similar neural substrates. For example, in the con-text of perceptual decision-making, where decision difficulty is varied by changing the uncertainty associated with a sensory stimulus, we would expect that regions which support sensory processing, such as visual motion area MT+ and lateral intra-parietal sulcus (LIP)[8,23], are also recruited when computing confidence in others' decisions. However, sensory representations on their own are unlikely to be sufficient because they do not reflect the characteristics of others' sensory systems – that is, others' sensory noise. Instead, the second component to a social confidence computation, the estimation of others' ability, may be provided by the so-called theory of mind (ToM) network. This ToM network includes temporoparietal junction (TPJ) and dor-somedial prefrontal cortex (dmPFC)[24–26] and is believed to support the representation of others' attributes as distinct from one's own. In support of this prediction, human fMRI has indi-cated that both TPJ and dmPFC are involved in the maintenance of running estimates of others' task performance[21,22,27].

Here, we develop an experimental paradigm to identify the components of a social confidence computation, and we combine behavioural analysis with computational modelling to show that people take into account both decision difficulty and others' ability. More specifically, we asked participants to place bets (post-decision wagers) on decisions on a random dot motion task[28] made by either themselves (self-trials) or one of three other players (other-trials) who differed in terms of task performance. To maximise reward on other-trials, participants should combine an estimate of decision difficulty with distinct estimates of the other players' ability. In this way, participants can predict that a good player may get a difficult stimulus wrong and that a bad player may get an easy stimulus right. We refer to this account as the ToM-model and we show that it provides a better fit to participants' post-decision wagers than alternative models which only track decision difficulty or others' ability. Analysis of fMRI data acquired during task behaviour indicated that the combi-nation of distinct estimates of decision difficulty and others' ability was associated with an interaction between brain systems involved in decision-making (LIP) and ToM (TPJ and dmPFC). Our results reveal a neurocomputational interplay between self- and other-related processes when forming confidence about others' decisions.

## Results

**Experimental paradigm**. Participants ($n = 21$) were asked to place post-decision wagers (PDWs) on either their own or another player's choices on a random dot motion task while undergoing fMRI (Fig. 1). At the start of a trial, a screen indicated whether they had to perform a self-trial or an other-trial. On both trial types, participants viewed a field of moving dots inside a circular aperture. On each update of the motion stimulus, a fraction of dots moved coherently to the left or the right (range: 0.005–0.5), whereas the remainder moved randomly. On self-trials, participants had to actively decide whether the average direction of dot motion was to the left or the right. On other-trials, participants watched the motion stimulus while the other player decided; participants were not told which decision the other player made.

On both trial types, a PDW screen then appeared requiring participants to choose between a safe and a risky option. The safe option delivered a small but guaranteed reward (range: 5–20 points). In contrast, the risky option delivered a greater reward if the self- or other-choice was correct and a corresponding loss if incorrect (range: 25–50 points). Each PDW thus provided an incentive-compatible metric of participants' confidence in a choice being correct. Finally, participants received feedback about the PDW outcome; participants could infer the accuracy of the self- or other-choice from this feedback.

Participants were paired in a block-wise manner with three other players who differed in their average choice accuracy (low: 0.55; medium: 0.75; high: 0.95). These other players were created using pilot data from participants who had performed the random dot motion task without a social component. Participants were told that the other players were not currently present and had made their decisions at another time. We highlight that our three main factors – motion coherence, others' ability and the reward difference between the risky and the safe option – were, by design, uncorrelated.

**Post-decision wagering on self- and other-trials**. Intuitively, the probability that participants select the risky option should depend on the confidence that a self- or other-choice is correct and the difference between the rewards available under the risky and the safe option. The higher the confidence, the higher the probability

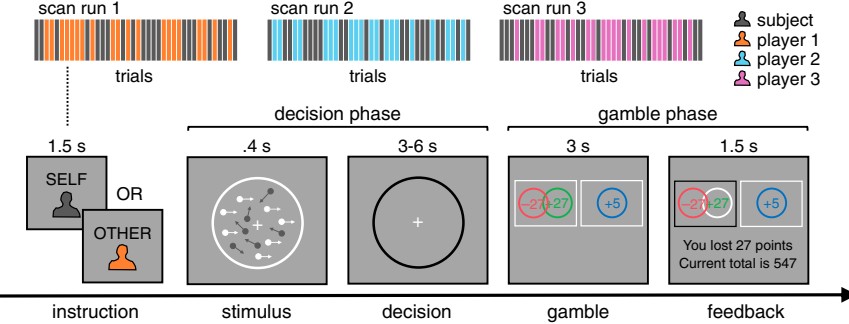

**Fig. 1 Social random dot motion task.** On each trial, participants placed a bet on a motion discrimination judgement (left or right) made by either themselves or one of three other players who varied in terms of their ability. The bet was made by choosing between a safe option, which yielded a small but guaranteed reward, and a risky option, which delivered a greater reward if the self- or other-choice was correct and a corresponding loss if incorrect. In this example, the participant chose the risky option (black box), but the self- or other-choice was incorrect. If participants chose the safe option, then they were informed what the outcome would have been had they selected the risky option. Participants were instructed how to calculate choice accuracy from the feedback screen. Participants were paired with the three other players in a block-wise manner, and the trial type was signalled at the start of a trial. For fMRI analysis, we refer to the time window from the onset of the motion stimulus to 3 s after the onset of the decision screen as the decision phase and the time window from the onset of the gamble screen to the offset of the feedback screen as the gamble phase.

that the risky option returns a reward, and the larger the reward difference, the higher the potential earnings from selecting the risky option. However, self- and other-trials should differ in terms of the factors that enter into a confidence computation: on both trial types, confidence should take into account the difficulty of the decision, but on other-trials, confidence should also take into account the ability of the other player.

To disentangle these contributions, we used a multiple logistic regression model to predict participants' trial-by-trial PDWs, with predictors including trial type (other = −0.5, self = 0.5), motion coherence, reward difference, others' ability (low = −0.5, medium = 0, high = 0.5) and the interactions between trial type and the three other terms (see Supplementary Fig. 1 for regression coefficients under full model and separate models for self- and other-trials). This analysis showed that the probability that participants selected the risky option increased with both motion coherence (Fig. 2a; coherence: $t(20) = 7.78$, $p < 0.001$) and reward difference (Fig. 2b; reward: $t(20) = 5.02$, $p < 0.001$), and that the impact of these factors did not depend on trial type (coherence x type: $t(20) = 0.18$, $p = 0.858$; reward x type: $t(20) = −0.44$, $p = 0.664$). The probability that participants selected the risky option also increased with others' ability (ability: $t(20) = 2.85$, $p = 0.010$), but this effect depended on trial type (type x ability: $t(20) = −4.60$, $p < 0.001$). Importantly, there was no effect of others' ability on self-trials (blue bars in Fig. 2c; ability under separate model for self-trials: $t(20) = −1.46$, $p = 0.161$), whereas on other-trials, the higher the ability of the other player, the higher the probability that participants selected the risky option (yellow bars in Fig. 2c; ability under separate model for other-trials: $t(20) = 4.52$, $p < 0.001$). Overall, participants were more likely to select the risky option on self- than other-trials (type: $t(20) = 4.47$, $p < 0.001$), with participants on average selecting the risky option on 72% of self-trials and 61% of other-trials.

Participants were not informed about the ability of the other players but instead had to learn it from experience. An obvious learning signal is the feedback about choice accuracy available at the end of each trial. In line with this hypothesis, participants were more likely to gamble on another player's choice when the other player's previous choice was correct compared to when it was incorrect (yellow bars in Fig. 2d; paired t-test: $t(20) = 2.97$, $p = 0.008$) – a history effect which was not present for self-trials (blue bars in Fig. 2d: paired t-test, $t(21) = −0.35$, $p = 0.733$).

Taken together, these results show that participants (1) were equally sensitive to decision difficulty and reward characteristics

on self- and other-trials and (2) tracked the ability of the other players and used this knowledge to guide their PDWs in a player-dependent manner. We note that there was no effect of others' ability on participants' task performance on self-trials, which we assessed using choice accuracy (logistic regression; coherence: $t(20) = 7.90$, $p < 0.001$; ability: $t(20) = −0.78$, $p = 0.444$) and reaction time (linear regression; coherence: $t(20) = −11.98$, $p < 0.001$; ability: $t(20) = 0.43$, $p = 0.674$).

**Computational model of confidence in self and others.** Qualitatively, the behavioural results fit with a ToM account in which the confidence in another player's decisions is based on both a self-related component, which reflects decision difficulty, and an other-related component, which reflects the ability of the other player. To test this interpretation formally, and to derive markers of the latent variables that underpin participants' behaviour for fMRI analysis, we compared candidate models of participants' PDWs.

In our task, participants should decide whether to gamble on a choice by comparing the expected value of the risky and the safe option, $\triangle EV = EV(\text{risky}) − EV(\text{safe})$. We modelled the mapping between $\triangle EV$ and gamble choices using a softmax function, $P(\text{gamble}) = \text{softmax}(\triangle EV)$. We allowed the parameters of this function to vary between trial types to accommodate individual variation in the tendency to gamble on choices made by oneself versus others. The expected value of the safe option is simply the amount offered, $EV(\text{safe}) = V(\text{safe})$. By contrast, the expected value of the risky option is computed by weighting the risky offer according to the confidence in a choice being correct, $EV(\text{risky}) = P(\text{correct}) \times V(\text{risky}) − [1 − P(\text{correct})] \times V(\text{risky})$. Here, we first consider the computation of confidence on self-trials, before turning to other-trials (see the Methods for mathematical details and Supplementary Table 1 for model parameters).

Building on earlier work on decision confidence[3,4], we modelled participants' confidence on self-trials using Bayesian decision theory (self-trials in Fig. 3). In particular, we assumed that participants represent the stimulus space as comprised of a set of distinct motion stimuli, each defined by a direction and a coherence. In the first version of the model, this representation consisted of linearly spaced motion stimuli. In a modified version, we allowed for an over-representation of low-coherence motion stimuli, consistent with a Weber-Fechner law in which the resolution of sensory perception diminishes for stimuli of greater

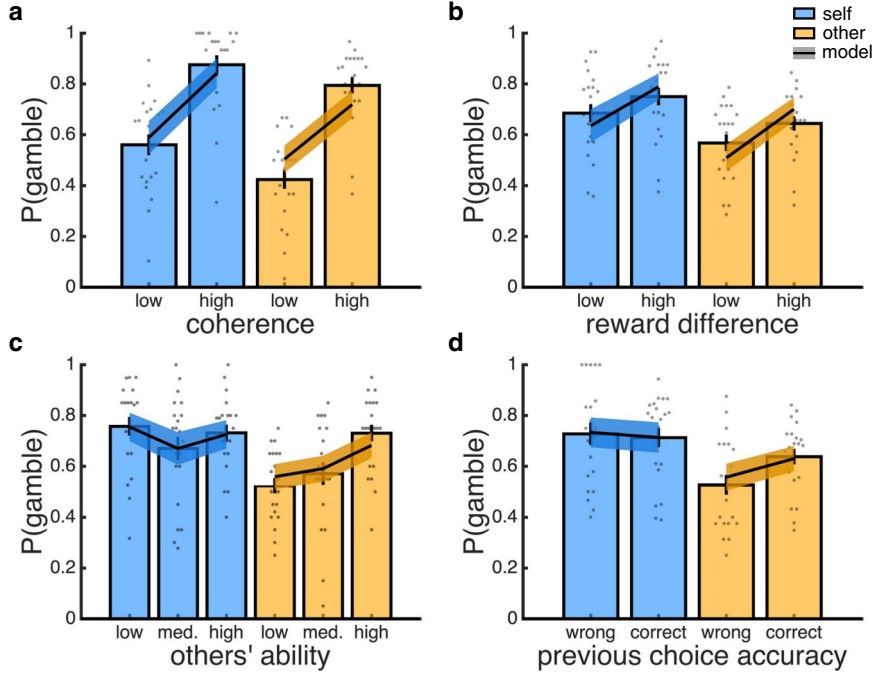

**Fig. 2 Behavioural results.** In each panel, data are split into self-trials (blue) and other-trials (yellow). **a** Probability gamble (i.e., proportion of trials in which the risky option was selected) as a function of coherence (median split). **b** Probability gamble as a function of the reward difference between the risky and the safe option (median split). **c** Probability gamble as a function of the ability of the other players. **d** Probability gamble as a function of choice accuracy on the previous trial of the same type. **a**–**d** Bar charts are empirical data, with individual participants overlaid as dots. Lines are data simulated under the best-fitting model (ToM-model). Empirical and simulated data are represented as group mean ± 95% CI, $n = 21$. Source data are provided as a Source Data file.

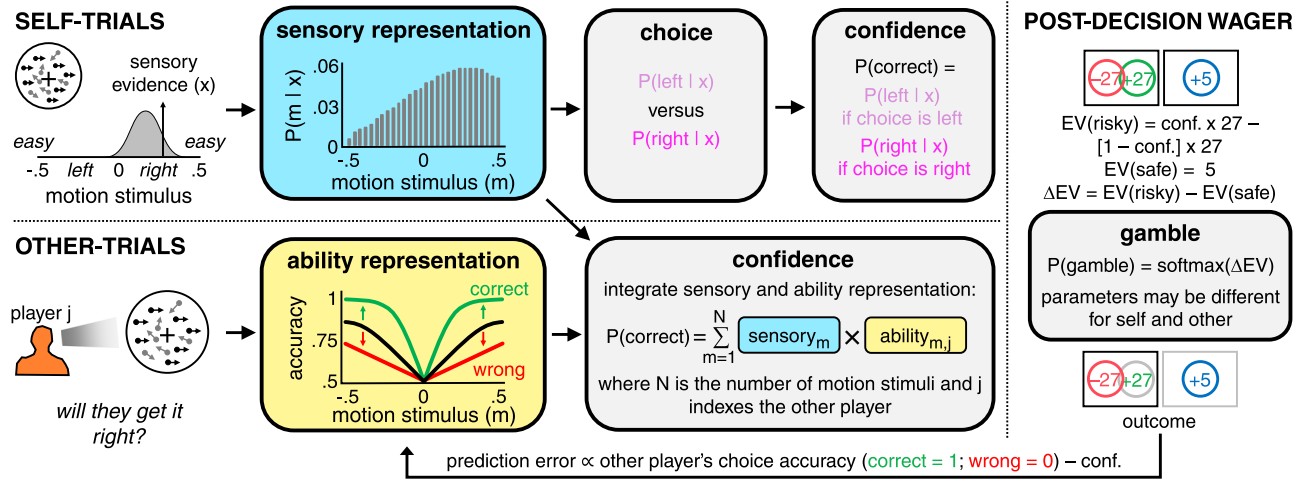

**Fig. 3 Illustration of a confidence computation for self and others under the ToM-model.** All models assumed that participants' choices and confidence were computed on self-trials according to Bayesian decision theory. Participants represent the stimulus space as comprised of a set of distinct motion stimuli, each defined by a direction (sign) and a coherence (absolute value). On each trial, participants receive sensory evidence sampled from a Gaussian distribution whose mean is given by the true motion stimulus and whose standard deviation is given by a participant's sensory noise. Participants compute a belief state over the stimulus space given the sensory evidence and their sensory noise (blue box) and use this belief state to generate both a decision about the motion direction and their confidence in this decision being correct. The ToM-model assumes that participants compute confidence in others' choices by combining their belief state over stimulus space with a representation of another player's expected accuracy for each motion stimulus (yellow box). The latter representation – in effect, a psychometric function – is derived from an estimate of the other player's sensory noise. This estimate is updated at the end of each trial using a Rescorla-Wagner learning rule that takes into account the accuracy of the other player's choice and the confidence in this choice being correct. As a result of this update, the psychometric function becomes steeper after a correct choice (green) and shallower after an incorrect choice (red). On both trial types, confidence is used to calculate the difference in expected value between the risky and the safe options. Finally, the difference in expected value is entered into a softmax function to obtain the probability of selecting the risky option. The softmax parameters were allowed to vary between self- and other-trials.

magnitude. On each trial, participants receive noisy sensory evidence, modelled as a sample from a Gaussian distribution centred on the true motion stimulus. Participants then compute a belief state over stimulus space – that is, the probability that each motion stimulus is shown on the current trial – given the sensory evidence and their own level of sensory noise. Finally, participants use this belief state to generate both a decision and their confidence in this decision being correct.

There are several ways in which a ToM account of a social confidence computation could be implemented. However, for consistency with earlier work on decision confidence[3,4], here we directly extended the Bayesian framework to the social case. This ToM-model assumes that participants compute confidence in the other player's decision by integrating their belief state over stimulus space with a distinct representation of the other player's expected accuracy for each motion stimulus (other-trials in Fig. 3). The ToM-model integrates these representations across the full stimulus space because participants do not know whether the other player decided left or right. The ability representation – equivalent to a belief about the other player's psychometric function – is derived from an estimate of the other player's sensory noise. This estimate is updated at the end of each trial according to the difference between the observed and the predicted success of the other player's choice — a social prediction error.

Under a ToM account, confidence on other-trials is influenced by both decision difficulty and others' ability. For completeness, and to rule out extraneous explanations of the behavioural results, we also considered models that only included one of these two components. First, we considered a self-projection model (S-model), which assumes that participants first ask themselves which choice they would have made and how confident they would have felt about this choice and then project this confidence estimate onto the other player. Under the S-model, confidence on other-trials is influenced by decision difficulty but not by others' ability. Second, building on previous work[22], we considered a performance-tracking model (Q-model), which assumes that participants use a Rescorla-Wagner rule to learn about the value of each player based on their historical choice accuracy and use this value as a proxy for confidence in their decisions. Under the Q-model, confidence on other-trials is influenced by others' ability but not by decision difficulty.

In keeping with the behavioural results, and in support of a ToM account of a social confidence computation, the ToM-model provided the best fit to participants' PDWs; the winning model included both a Weber-Fechner law and separate softmax parameters for self and other (see Supplementary Fig. 2 for model comparison). As shown in Fig. 2, the ToM-model captured all qualitative features of the behavioural data (see Supplementary Fig. 3 for predictions under the best-fitting version of each model class). In addition, the ToM-model captured the trial-by-trial evolution of participants' PDWs for each of the three other players (Supplementary Fig. 4).

We performed two control analyses to assess the robustness of our modelling approach. First, we performed a model identifiability analysis using the best-fitting version of each model class[29]. Specifically, we simulated data under each model and then fitted each model to these data. This analysis showed that the models were discriminable within the constraints of our experimental paradigm (Supplementary Fig. 5). Second, the models infer participants' true confidence from their PDWs – raising the possibility that risk and/or loss aversion may have biased the model comparison results and/or distorted the model-based estimates of confidence for subsequent fMRI analysis[30]. To rule out this possibility, we re-fitted the best-fitting version of each model class after adding a utility function, which transforms the

expected value of the gamble options into subjective utilities and allows for individual differences in risk and/or loss aversion[30]. This analysis confirmed the ToM-model as providing the best account of the behavioural data and showed that the model-based estimates of confidence remained the same when inferred with or without a utility function (Supplementary Fig. 6).

**Neural basis of a social confidence computation**. We next turned to the fMRI data acquired during task behaviour to identify neural substrates that may contribute to a social confidence computation. The behavioural and modelling results supported a ToM account in which participants compute confidence in another player's decisions by integrating distinct estimates of decision difficulty and others' ability. The first component, a representation of decision difficulty, is directly linked to the motion stimulus and in particular motion coherence: the higher the coherence, the easier the decision. The second component, a representation of others' ability, must be maintained separately from one's own ability and learned from feedback about the other player's choice accuracy. We would therefore expect brain systems traditionally involved in sensory and social processing to interact during a social confidence computation and in turn that the putative social areas carry learning signals relating to others' task performance.

To identify brain regions that were differentially activated by self- and other-related processing, we first estimated whole-brain contrasts between self- and other-trials during the decision and the gamble phases (Fig. 1). During the decision phase, the self > other contrast identified classic perceptual decision-making regions, including extrastriate cortex, posterior parietal cortex and cingulate cortex, and, in line with only self-trials requiring active choice, motor regions (Fig. 4). By contrast, the other > self contrast identified a classic ToM network, including middle temporal gyrus, TPJ and dmPFC (Fig. 4). During the gamble phase, the self > other contrast did not identify differential activity, whereas the other > self contrast again identified the ToM network (results not shown for gamble phase – see Supplementary Table 2 and Supplementary Table 3 for all clusters surviving whole-brain correction for decision and gamble phases).

We next focused on a subset of these brain regions to assess the neural evidence for a ToM account of a social confidence computation; these regions were defined using independent region of interest (ROI) masks (Fig. 5). First, we hypothesised that visual motion area MT+, identified in a separate localiser scan, and a human homologue of monkey LIP, encompassing posterior parts of the superior parietal lobule and the intraparietal sulcus[31], may support a sensory representation of the motion stimulus which informs confidence in both one's own and others' choices. On this account, MT+ and LIP, implicated in sensory integration on the random dot motion task[8,32–35], should encode motion coherence on both trial types. We surmise that the higher baseline activity in MT+ and LIP on self-trials (Fig. 4) was driven by decision processes specific to self-trials such as the requirement for making an active choice. Second, we hypothesised that TPJ and dmPFC, implicated in social inference[21,22,27], may combine a sensory representation of the motion stimulus with a distinct representation of others' ability as required by a social confidence computation. On this account, TPJ and dmPFC should track confidence in others' choices after controlling for the sensory representation of motion coherence. We tested these predictions using a complementary analysis approach: we (1) visualised temporally resolved neural encoding profiles by applying sliding-window regressions to up-sampled single-trial ROI activity time courses and (2) then estimated single-trial canonical hemodynamic response functions (c-HRF) for significance testing.

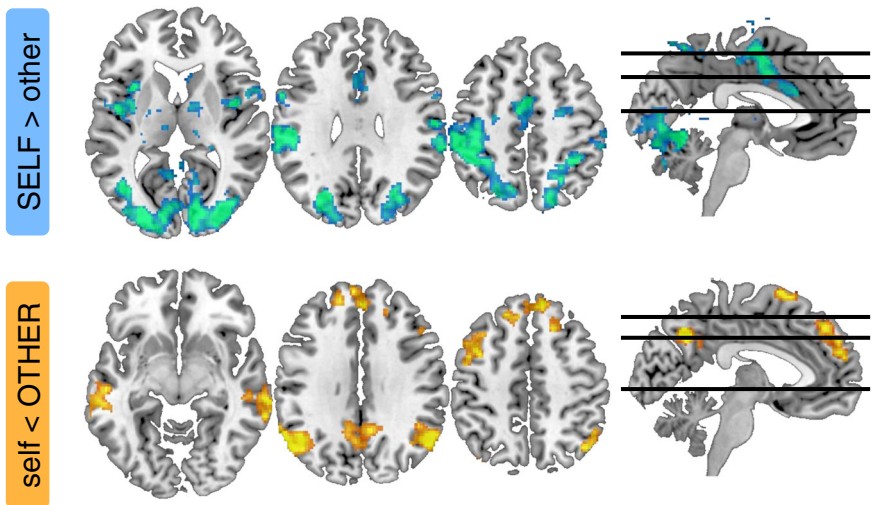

**Fig. 4 Whole-brain activations for self- and other-related processing during decision phase.** Images display clusters surviving whole-brain correction ($p < 0.05$, FWE-corrected for multiple comparisons at a cluster-defining threshold of $p < 0.001$, uncorrected) for the contrast between self- and other-trials during the decision phase (cold: SELF > other; warm: self < OTHER). Images are shown at $p < 0.001$, uncorrected. All clusters surviving whole-brain correction during the stimulus and gamble phases are detailed in Supplementary Tables 2 and 3.

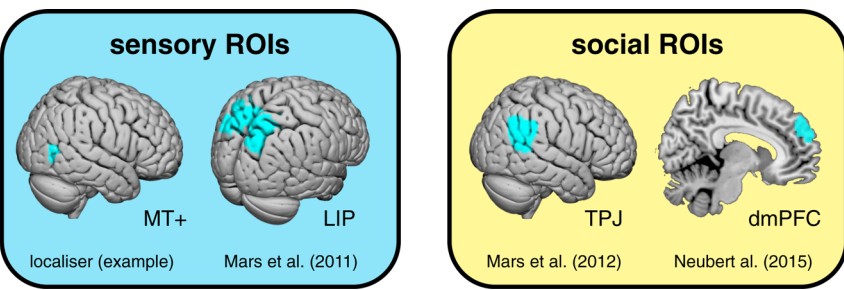

**Fig. 5 Anatomical masks for ROI analyses.** The MT+ mask was specified for each participant using a localiser scan. The other masks were specified using published connectivity-based parcellation atlases (see Methods).

A ToM account predicts that regions supporting the formation of a sensory representation of the motion stimulus should also contribute to a social confidence computation. In other words, we would expect our sensory ROIs to carry information about motion coherence on both self- and other-trials. To test this hypothesis, we quantified the neural impact of trial type (social; now coded as self = −0.5 and other = 0.5), motion coherence and their interaction. As expected under a ToM account, MT+ and LIP tracked motion coherence on both trial types (Fig. 6). In MT+, the response profile was the same on self- and other-trials: the higher the coherence, the higher the activity (c-HRF regression; social: $t(20) = -7.72$, $p < 0.001$; coherence, $t(20) = 5.42$, $p < 0.001$; interaction, $t(20) = -0.85$, $p = 0.408$; coherence on other-trials, $t(20) = 3.94$, $p < 0.001$; coherence on self-trials, $t(20) = 3.65$, $p = 0.002$). In contrast, the LIP response profile differed between trial types: the higher the coherence, the higher the activity on other-trials, but the lower the activity on self-trials (c-HRF regression; social: $t(20) = -7.86$, $p < 0.001$; coherence, $t(20) = 0.20$, $p = 0.844$; interaction, $t(20) = 3.07$, $p = 0.006$; coherence on other-trials, $t(20) = 2.94$, $p = 0.008$; coherence on self-trials, $t(20) = -2.14$, $p = 0.045$).

The LIP response pattern is consistent with previous observations of posterior parietal fMRI activity during active choice (characteristic of self-trials) versus passive viewing (characteristic of other-trials). When participants are required to make an active choice, posterior parietal activity has been reported to decrease with motion coherence[34]. In contrast, during passive viewing, posterior parietal activity has been found to increase with motion

coherence[36]. Such a flip is what we would expect if LIP neurons integrate sensory information into a belief state over stimulus space – equivalent to the sensory representation in our computational models. During active choice, the sensory integration process terminates earlier for high-coherence than low-coherence motion stimuli[23] and bulk neural activity is therefore likely to be lower for high-coherence than low-coherence motion stimuli. In contrast, during passive viewing, such early termination of a choice process does not occur and motion coherence itself is likely to drive bulk neural activity. In support of this explanation, previous work has shown that, when motion coherence and reaction time are dissociated during active choice, posterior parietal activity increases with motion coherence, just as observed on other-trials[8].

Having found that MT+ and LIP track motion coherence on both trial types, we next turned to the social ROIs in order to assess their roles in a social confidence computation. The whole-brain analysis showed that the ToM network – including TPJ and dmPFC – was more active on other- than self-trials during both trial phases. However, a ToM account predicts that this network should not only discriminate between self and other but also differentially encode confidence for self and other. While previous work suggests that TPJ should be selective for a social confidence computation[37,38], the potential role of dmPFC is less clear; recent research indicates that dmPFC does not selectively encode social information but instead supports a separation between self- and other-related information when required by the task[22,39,40]. To test these hypotheses, we quantified the neural impact of trial type

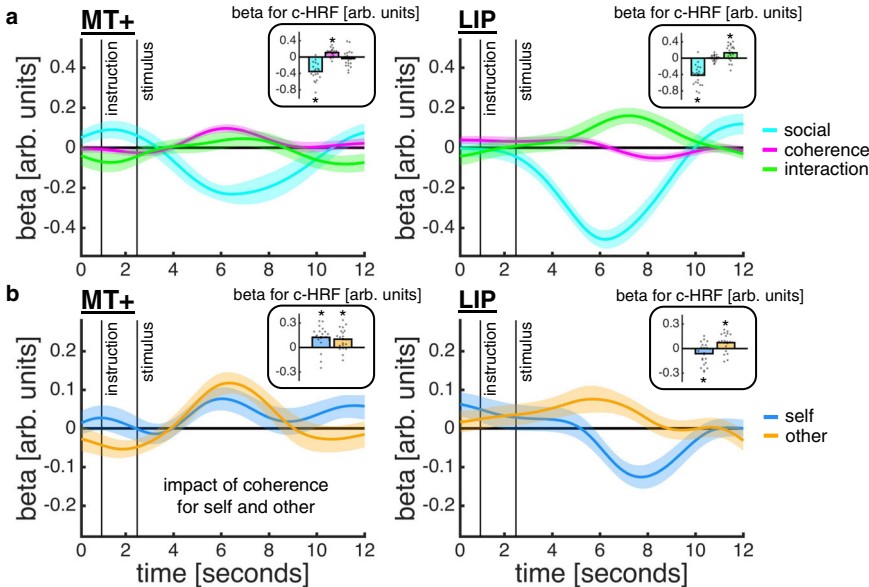

**Fig. 6 Encoding of motion coherence in MT+ and LIP. a** The time courses are coefficients from a regression in which we predicted z-scored MT+ (left) and LIP (right) activity time courses using trial type (cyan; other = 0.5; self = −0.5), z-scored motion coherence (pink) and their interaction (green). The insets show coefficients from an analysis of canonical HRFs (c-HRFs; asterisk indicates statistical significance, $p < 0.05$, one-sample $t$-test against zero). **b** Same approach as in **a**, but now quantifying the impact of motion coherence separately for each trial type. **a–b** Data are represented as group mean ± SEM, $n = 21$. Source data are provided as a Source Data file.

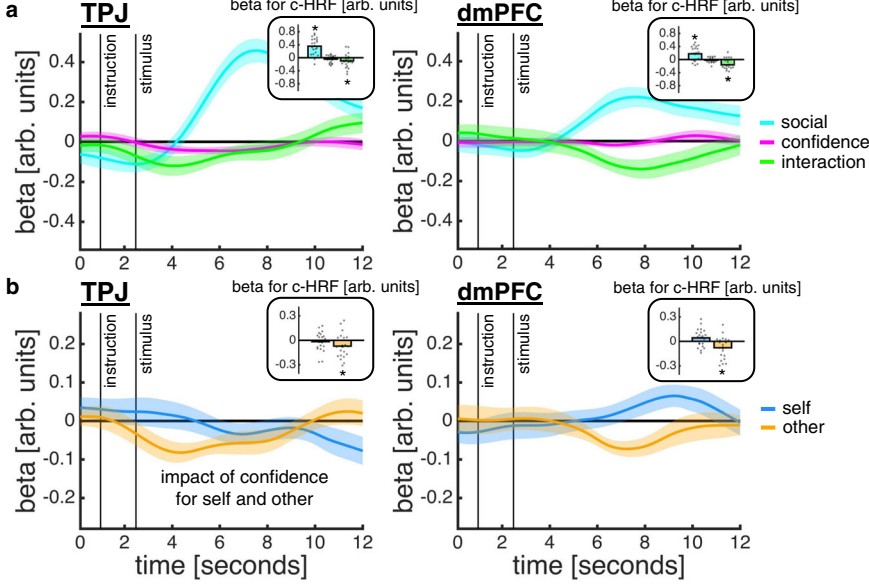

**Fig. 7 TPJ and dmPFC support a social confidence computation. a** The time courses are coefficients from a regression in which we predicted z-scored TPJ (left) and dmPFC (right) activity time courses using trial type (cyan; other = 0.5; self = −0.5), z-scored model-based confidence estimates as computed under the ToM-model (pink; first orthogonalised with respect to motion coherence) and their interaction (green). The insets show coefficients from an analysis of canonical HRFs (c-HRFs; asterisk indicates statistical significance, $p < 0.05$, one-sample $t$-test against zero). **b** Same approach as in **a**, but now quantifying the impact of model-based confidence estimates separately for each trial type. **a–b** Data are represented as group mean ± SEM, $n = 21$. Source data are provided as a Source Data file.

(social; self = −0.5 and other = 0.5), our model-based confidence estimates (as computed under the ToM-model for both self and other) and the interaction between these terms. To control for low-level sensory effects, we orthogonalised the model-based confidence estimates with respect to motion coherence.

This analysis showed that both TPJ and dmPFC encoded an interaction between trial type and the model-based confidence estimates (Fig. 7a; c-HRF regression; TPJ social: $t(20) = 6.33$, $p < 0.001$; TPJ confidence, $t(20) = −1.79$, $p = 0.089$; TPJ

interaction, $t(20) = −2.10$, $p = 0.049$; dmPFC social: $t(20) = 3.70$, $p = 0.001$; dmPFC confidence, $t(20) = −1.15$, $p = 0.264$; dmPFC interaction, $t(20) = −3.44$, $p = 0.003$). However, the nature of the interaction effect differed between regions (Fig. 7b). In support of selectivity for social inference, TPJ encoded the model-based confidence estimates on other-trials only, with higher activity when confidence was low (c-HRF regression; confidence on other-trials, $t(20) = −2.17$, $p = 0.043$; confidence on self-trials, $t(20) = −0.60$, $p = 0.555$). By contrast, dmPFC encoded model-based

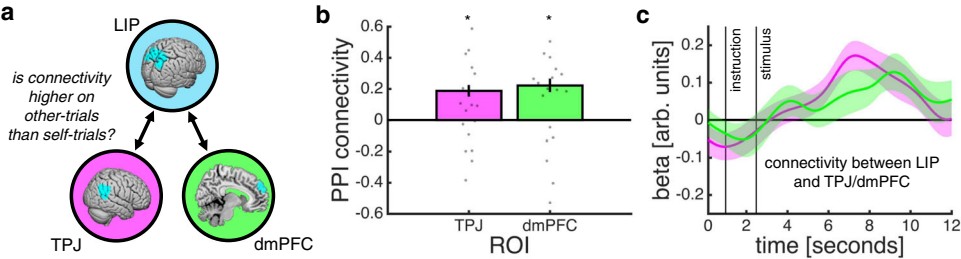

**Fig. 8 Functional coupling between sensory and social ROIs during a social confidence computation. a** Schematic of PPI analysis testing whether the correlation between LIP activity (seed region) and TPJ/dmPFC activity is higher on other- than self-trials. **b** Contrast estimates from PPI analysis (other > self) as implemented by the Generalised PPI toolbox (asterisk indicates statistical significance, $p < 0.05$, one-sample $t$-test against zero). **c** Visualisation of activity time courses driving effects documented in **b**. The time courses are coefficients from a regression in which we predicted z-scored TPJ/dmPFC activity time courses using an interaction between z-scored LIP activity time courses and trial type (self = $-0.5$; other = $0.5$), while controlling for the main effect of each term. **b–c** Data are represented as group mean ± SEM, $n = 21$. Source data are provided as a Source Data file.

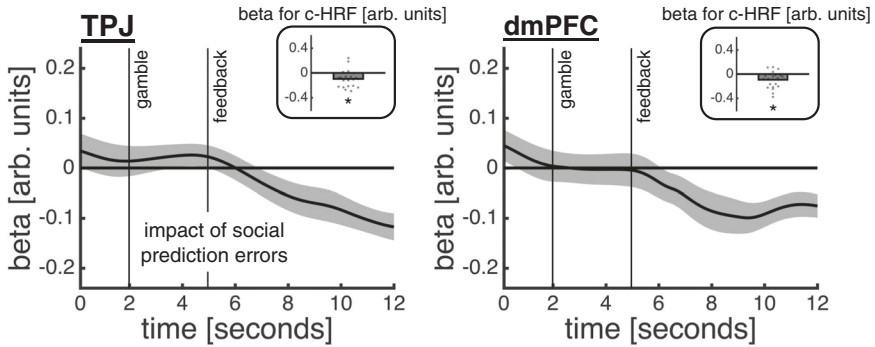

**Fig. 9 Encoding of social prediction errors in TPJ and dmPFC.** The time courses are coefficients from a regression in which we predicted z-scored TPJ (left) and dmPFC (right) activity time courses using z-scored social prediction errors as computed under the best-fitting ToM-model. The insets show coefficients from an analysis of canonical HRFs (c-HRFs; asterisk indicates statistical significance, $p < 0.05$, one-sample $t$-test against zero). Data are represented as group mean ± SEM, $n = 21$. Source data are provided as a Source Data file.

confidence estimates on both trial types: activity decreased with confidence on other-trials, but increased with confidence on self-trials, although the effect on self-trials did not reach statistical significance (c-HRF regression; confidence on other-trials, $t(20) = -2.67$, $p = 0.015$; confidence on self-trials, $t(20) = 1.79$, $p = 0.089$). The net effect of this inverse response profile in dmPFC is a larger neural distinction between other- and self-trials when confidence is low compared to when confidence is high.

Having established that TPJ and dmPFC track confidence estimates about others' choices, we next turned to the within- and across-trial dynamics of a social confidence computation. We reasoned that, if the ToM network receives a sensory representation of the motion stimulus from classic perceptual decision-making regions and in turn furnishes this sensory representation with a social representation of another player's ability, then functional connectivity between our sensory and social ROIs should be higher on other- than self-trials. To test this hypothesis, we performed a psychophysiological interaction (PPI) analysis using MT+ or LIP as seed region and trial type as the psychological variable (Fig. 8a). In line with a ToM account, the PPI analysis revealed that connectivity between LIP and TPJ/dmPFC was higher on other- than self-trials (Fig. 8b, c; one-sample $t$-test: TPJ, $t(20) = 2.50$, $p = 0.021$; dmPFC, $t(20) = 2.56$, $p = 0.019$). These relationships were not found for MT+ (one-sample $t$-test: TPJ, $t(20) = 0.51$, $p = 0.615$; dmPFC, $t(20) = -0.80$, $p = 0.432$).

Finally, having examined how sensory and social regions interact to compute confidence in others' choices, we analysed how the representation of another player's ability is updated with

task experience. The behavioural and modelling results indicated that participants revised their estimate of the other player's sensory noise based on the difference between the accuracy of the other player's choice and their confidence in this choice being correct (see schematic of learning mechanism in Fig. 3). Consistent with a role in supporting this estimate, both TPJ and dmPFC tracked social prediction errors as computed under the ToM-model (Fig. 9; c-HRF regression; TPJ, $t(20) = -3.18$, $p = 0.005$; dmPFC, $t(20) = -3.12$, $p = 0.005$). We note that these response profiles remained after orthogonalising the social prediction error with respect to either the outcome term (accuracy) or the prediction term (confidence).

## Discussion

Computing confidence in others' decisions is critical for success on a range of competitive and cooperative tasks. While previous research has examined how we track others' ability, little is known about whether, or how, we combine this knowledge with an assessment of the decision problem at hand to form a confidence estimate. In many situations, simply knowing someone else's ability is insufficient for an accurate confidence computation as the difficulty of a decision may vary substantially. For example, an excellent medical doctor may not be able to diagnose a complex case, whereas someone who is generally bad at mathematics may still be able to solve a simple arithmetical problem. In situations like these, distinct estimates of others' ability and decision difficulty are needed in order to predict whether someone else will be successful at a given moment in time.

Here, we addressed how people solve this problem using a perceptual decision task in which participants placed bets on choices about stimuli of varying difficulty made by either themselves (self-trials) or one of three other players (other-trials) who differed in terms of their ability. By combining behavioural analysis with computational modelling, we found that participants tracked the ability of each player and combined this knowledge with an estimate of decision difficulty in order to decide whether to gamble on the other players' choices being correct. Using fMRI, we found that a social confidence computation was supported by an interaction between brain systems involved in perceptual decision-making (MT+ and LIP) and social cognition (TPJ and dmPFC): MT+ and LIP tracked decision difficulty on both self- and other-trials; coupling between LIP and TPJ/dmPFC increased on other-trials; and TPJ/dmPFC tracked confidence in the other players' choices after controlling for decision difficulty and tracked prediction error learning about the ability of the other players. Taken together, these results indicate that TPJ/dmPFC augment a representation of decision difficulty provided by LIP with a representation of the other players' ability.

Our winning model, the ToM-model, is a natural extension of Bayesian models of confidence from the individual to the social case[4]. In this model, decision difficulty and others' ability are taken into account by integrating a belief state over stimulus space with a representation of the other players' expected accuracy for each stimulus. Our goal was to test whether a social confidence computation takes into account both decision difficulty and others' ability, rather than seeking to arbitrate between alternative algorithms that may support this computation. For example, people may solve the problem in a more model-free manner, by first calculating the confidence in the decision that they themselves would have made and then adjusting this estimate by a scalar value that reflects the performance of the other player (e.g., a combination of the S- and Q-models). Both this heuristic account and the ToM-model embody a hypothesis that a social confidence computation involves self- and other-related components, but they make subtly different predictions (e.g., the difference in confidence for a low-ability and high-ability player depends on the level of motion coherence under the ToM-model but is constant across levels of motion coherence under this more model-free alternative). Future studies are needed to disambiguate these related accounts (e.g., extensive psychophysical mapping of the relationship between motion coherence and confidence in others' choices).

Our confidence models were all grounded in Bayesian decision theory on self-trials, but they differed in their solution to a social confidence computation on other trials. The Q-model solves this problem in a model-free manner, using running estimates of the other players' choice accuracy as a proxy for confidence in their choices. In contrast, the best-fitting model, the ToM-model, solves the problem in a Bayesian manner, by integrating distinct representations of decision difficulty and others' ability. We acknowledge that the computational basis of (non-social) confidence is still debated, with different studies supporting either Bayesian[4,8,41–43] or non-Bayesian[44–46] accounts. However, the grounding of our confidence models in Bayesian decision theory is unlikely to have biased our results. In keeping with standard signal detection theory, we modelled the sensory evidence as a single sample from a Gaussian distribution. Under these constraints, Bayesian and non-Bayesian confidence estimates are monotonically related and will therefore make similar predictions[3].

The question of how people infer what others think, feel or intend, and the relationship between inference about one's own and others' internal states, has attracted widespread attention in philosophy, psychology and neuroscience[26,47,48]. Our findings are in line with a view that self- and other-inference are distinct processes that involve distinct representations of oneself and others[24,49] – and go against a view that other-inference relies on self-projection[50,51] or that self-inference co-opts processes for inference about others[48]. At a behavioural level, we found that a social confidence computation involves distinct self- and other-related components – people did not simply simulate which choice they would have made and how confident they would have felt about this choice. At a neural level, we found that self- and other-related processes activated distinct brain systems. In further support of this neural dissociation, a repetition suppression analysis also indicated that LIP was differentially activated by self-related processes, whereas TPJ and dmPFC were differentially activated by other-related processes (Supplementary Fig. 7). In particular, the higher level of LIP activity on self-trials was suppressed on self-trials preceded by self-trials as compared to self-trials preceded by other-trials, whereas the higher level of TPJ and dmPFC activity on other-trials was suppressed on other-trials preceded by other-trials as compared to other-trials preceded by self-trials. Under the repetition suppression framework, such neural attenuation occurs when information for which neuronal populations are selective is repeatedly represented in close temporal succession[52].

Our study characterises the neurocomputational basis of a common social problem – how we can make predictions about the choice accuracy of another system than oneself. As is true for most social neuroscience studies, it is possible that one could devise an analogous task where the other system is not human but artificial (e.g., a motion detection algorithm) and find that this task recruits similar neurocomputational mechanisms. However, this result would not alter the importance of these neurocomputational mechanisms for understanding human social behaviour. While our study was not designed to address the issue of social uniqueness[53], the involvement of a ToM network in the computation of confidence in other players' choices supports that our task taps into fundamental social processes. In addition, the recruitment of a ToM network supports our interpretation that participants computed social confidence by estimating the other players' sensory noise – a process akin to inference about latent mental states which similarly can explain variability in others' behaviour – and not simply by averaging trial-by-trial feedback about the other players' choices.

More broadly, our study extends our understanding of the neural basis of social cognition. TPJ and dmPFC have previously been shown to support learning of summary statistics that reflect others' ability on a variety of tasks, including the reliability of advice on a bandit task[27], recency-weighted performance on a reaction-time task[22] or prediction accuracy on a stock-market task[21]. Our results show that TPJ and dmPFC also support the integration of such knowledge about others' ability with information about the current task state – a process that facilitates accurate predictions about others under conditions of varying task difficulty. There were, however, subtle differences between the activity profiles of TPJ and dmPFC. In line with a hypothesis that TPJ is selective for social inference, TPJ tracked the model-based confidence estimates only on other-trials. By contrast, dmPFC tracked these estimates on both trial types but in an inverse manner – resulting in a larger neural distinction between self and other when confidence was low. This result fits with evidence that dmPFC supports a separation between self- and other-related information in social contexts[22,39,40].

While TPJ involvement was restricted to other-evaluation in the current study, it remains plausible that TPJ may be involved in self-evaluation in other situations. On other-trials, but not self-trials, participants had to learn novel mappings between stimulus

space and choice accuracy. In addition, unlike on self-trials, participants did not observe the other player's choice and had to consider that either choice could have been made. It remains to be seen whether self-evaluation co-opts putative social brain areas in situations where one's own ability needs to be learned across time[48] or where counterfactual thinking is required[54].

In summary, our study shows (1) that people form confidence in others' decisions by flexibly combining knowledge about others' ability with an assessment of the decision problem at hand and (2) that this process is supported by an interaction between brain systems traditionally involved in decision-making and social cognition. We highlight, however, that our study only addresses one of the two sides of a social confidence computation. In many cases, adaptive control of social behaviour requires not only that we can predict the success of others' choices – as examined here – but also that we can infer how confident others themselves feel about these choices. For example, in a strategic game like chess, our response to an opponent's mistake may depend on whether we think that they themselves realise that they have erred. How our confidence in others interacts – in a potentially reciprocal manner – with inference about others' confidence awaits further study.

## Methods

**Participants**. Participants performed a random dot motion task in a prescan and a scan session conducted on the same day. Twenty-two adults with no history of psychiatric or neurological disorders took part in the study. One participant was excluded due to poor performance in the pre-scan session, leaving twenty-one participants for analysis (12 female, mean ± SD age = 22.6 ± 4.4 years). Participants provided informed consent, and the study was approved by New York University's University Committee on Activities Involving Human Subjects. Participants received a show-up fee of US\$15, an additional US\$25 for completing the MRI scan and were informed they could earn an additional performance-based bonus (in reality, all participants received a US\$9 bonus).

## Experimental paradigm

*Random dot kinematograms*. Participants viewed random dot kinematograms (RDKs) contained in a circular aperture (7 degrees in diameter). Each RDK was made up of three independent sets of dots (each dot was 0.12 degrees in diameter) shown in consecutive frames. Each set of dots were shown for one frame (about 16 ms) and then replotted again three frames later (about 50 ms). Each time a set of dots was replotted, a subset of the dots, determined by the motion coherence, $\theta$, was displaced in the direction of motion at a speed of 6 degrees s$^{-1}$, whereas the rest of the dots were displaced at random locations within the aperture. The motion direction, $k$, was to the left or to the right along the horizontal meridian. The use of three sets of dots means, for example, that dot positions in frame one are correlated with dot positions in frame four, seven and so forth. The dot density was fixed at 30 dots degrees$^{-2}$ s$^{-1}$.

*Pre-scan session: familiarisation*. Participants were familiarised with the basic random dot motion task prior to the scan session. Each trial began with the presentation of a fixation cross at the centre of a circular aperture. After a fixed delay (0.5 s), participants viewed an RDK (0.4 s). Once the RDK had terminated, participants were asked to indicate whether the net direction of dot motion was to the left or to the right along the horizontal meridian. In particular, participants were presented with a white box on the left and a white box on the right, corresponding to the two choice options. Once a choice had been made, the outline of the chosen option turned black (0.5 s), and participants received feedback about choice accuracy (low-pitched tone: error; high-pitched tone: correct). If participants did not respond within 4 s, then the choice was scored as an error. The coherence, $\theta$, was taken from the set, $\Theta \in \{0.03, 0.06, 0.12, 0.24, 0.48, 1\}$, counterbalanced across trials. The motion direction, $k$, was sampled randomly from left or right. Participants completed 240 trials.

*Scan session: self-other*. Participants performed the same random dot motion task again, except that they were now also required to place post-decision wagers (PDWs) on the accuracy of either their own or another player's choices.

Each trial began with the presentation of a fixation cross at the centre of a circular aperture (1.5 s). Participants then viewed a message instructing either a 'self-trial' or an 'other-trial' (1.5 s). On both trial types, after a brief delay (0.5 s), participants viewed an RDK (0.4 s). After the RDK had terminated, participants entered the choice screen, with the fixation cross and the circular aperture remaining visible. On self-trials, participants actively decided about the net direction of dot motion. On other-trials, participants passively viewed the screen

while the other player made a decision about the motion stimulus; participants were not informed whether these decisions were left or right. On both trial types, the circular aperture changed its colour to black once a choice had been made. The other player's choice reaction time was sampled uniformly from the range 0.5–1 s. On self-trials, if participants did not respond within 3 s, then the trial was aborted – participants had to wait for the duration that a full trial would have taken before proceeding to the next trial. The delay between the offset of the RDK and the offset of the choice screen was sampled uniformly from the range 3–6 s.

Participants were then presented with a PDW screen (3 s) containing a safe and a risky option, indicated by a white box on each side of the screen (random assignment). The safe option, indicated by a blue circle, delivered a small but certain reward regardless of choice accuracy. The risky option delivered a larger reward, indicated by a green circle, if the self- or other-choice was correct, and a corresponding loss, indicated by a red circle, if the self- or other-choice was wrong. The safe value was drawn uniformly from the range 5 to 20 points, and the risky value was drawn uniformly from the range 25 to 50 points. Once participants had made a PDW, they were presented with a feedback screen (1.5 s). The screen indicated the chosen PDW option (relevant box highlighted in black) and the accuracy of the choice (if correct, the green circle maintained its colour, whereas the red circle turned white, and if incorrect, the red circle maintained its colour, whereas the green circle turned white). In this way, participants were informed about the outcome on the current trial as well as the outcome that would have been obtained had they chosen the other PDW option. In addition, the screen also indicated the total earnings accumulated up to the current trial. Participants started the scan session with 500 points and were told that their performance-based bonus depended on the total amount of points accumulated across the experiment. If participants did not make a PDW within 3 s of the PDW onset, then they lost the points associated with the safe option.

Participants performed 3 blocks (scan runs) consisting of 40 trials, with 20 self-trials and 20 other-trials randomly interleaved (120 trials in total). On each block, participants were paired with a new player. All participants faced the same three players. We created the three players using a pilot dataset of random dot motion decisions obtained from previous participants who had visited the lab. We identified three pilot participants who on average achieved about 60% (low ability), 75% (medium ability) and 90% (high ability) choice accuracy and then sub-selected 20 trials, which covered a range of coherences (0.001–0.5), had an even split of leftwards and rightwards motion and satisfied the above requirements for choice accuracy. On each block, the same set of motion stimuli were shown on self-trials and other-trials but in a randomised order. The order of the three players across blocks was randomised across participants.

## FMRI

*Procedure*. The scan session consisted of 5 scan runs. First, we acquired structural images. Second, participants performed the self-other task over 3 runs ($3 \times 40 = 120$ trials; 252 volumes per run). Lastly, we ran a localiser scan in which participants viewed alternating displays (12 s) of static and coherently moving dots ($2 \times 10 = 20$ displays; 215 volumes).

*Acquisition*. MRI data were acquired on a 3 T Siemens Allegra scanner at New York University's Center for Brain Imaging. T1-weighted structural images were acquired using a 3D MPRAGE sequence: $1 \times 1 \times 1$ mm resolution voxels; 176 sagittal slices. BOLD T2*-weighted functional images were acquired using a Siemens epi2d BOLD sequence: $3 \times 3 \times 3$ mm resolution voxels; 42 transverse slices; $64 \times 64$ matrix; TR = 2.24 s; TE = 30 ms; slice tilt = −30 degrees T > C; slice thickness = 3 mm; interleaved slice acquisition. Local field maps were recorded for distortion correction of the acquired EPI data.

*Pre-processing*. MRI data were pre-processed using SPM12 (Wellcome Trust, London) The first 5 volumes of each functional run were discarded to allow for T1 equilibration. Functional images were slice-time corrected, re-aligned and un-warped using the collected field maps[55]. Structural T1-weighted images were co-registered to the mean functional image of each participant using the iterative mutual-information algorithm. Each participant's structural image was segmented into grey matter, white matter and cerebral spinal fluid using a nonlinear deformation field to map it onto a template tissue probability map[56]. These deformations were applied to both structural and functional images to create new images spatially normalised to the Montreal Neurological Institute (MNI) space and interpolated to $2 \times 2 \times 2$ mm voxels. Normalized images were spatially smoothed using a Gaussian kernel with full-width half-maximum of 8 mm. Motion correction parameters estimated from the re-alignment procedure and their first temporal derivatives (12 regressors in total) were included as confounds in the first-level analysis for each participant.

## Quantification and statistical analysis

*Trial exclusion*. We excluded trials when a decision was not made during the choice window or when a gamble was not made during the gamble window.

*Behavioural analysis*. We used multiple logistic regression to predict participants' PDWs. We performed separate regressions for each participant, and then tested for

group-level significance by comparing coefficients pooled across participants to zero ($p < 0.05$, one-sample $t$-test). We z-transformed all continuous predictors.

*Computational models of confidence.* On each trial, a participant is asked to make a post-decision wager (PDW) about either their own choice or the choice of another player. On self-trials, the participant actively makes a choice about the motion stimulus. On other-trials, the participant is presented with the motion stimulus but not the other's choice. The participant has to decide between a risky and a safe option, by combining the option values with an estimate of the probability that the choice is correct. We first consider models of PDWs on self-trials, before turning to other-trials. The models are grounded in Bayesian decision theory[4,57].

Sensory sample: On each trial, a participant, $s$, receives a sensory sample, $x$, randomly sampled from a Gaussian distribution, $x \in N(k\theta_m, \sigma_s)$, where $m \in M = \{1, 2, \dots, n\}$, $\theta \in \Theta = \{\theta_1, \theta_2, \dots, \theta_n\}$ is the motion coherence, $k \in K = \{-1, 1\}$ is the motion direction ($-1$: left; $1$: right), $n$ is the number of levels of coherence and $\sigma_s$ is the participant's level of sensory noise.

Self-trials: The participant represents the task as comprised of a set of states $(k, m)$, corresponding to the set of possible motion stimuli defined by direction and coherence, $k\theta_m$. The participant computes a belief distribution over states given their sensory sample, $x$. The probability of each state conditional on $x$ is given by:

$$P(k, m|x) = \frac{P(x|m, k)P(m, k)}{\sum_{k'}\sum_{i=1}^{n} P(x|m_i, k')P(m_i, k')} = \frac{P(x|m, k)}{\sum_{k'}\sum_{i=1}^{n} P(x|m_i, k')} \quad (1)$$

Note that we relied on the fact that all states are equally probable by design (with probability $1/2n$). The likelihood of $x$ conditional on a state is given by:

$$P(x|k, m) = \varphi(x; k\theta_m, \sigma_s) \quad (2)$$

where $\varphi$ is the normal probability density function evaluated at $x$:

$$\varphi(x; \mu, \sigma) = \frac{1}{\sigma\sqrt{2\pi}}e^{\frac{-(x-\mu)^2}{2\sigma^2}} \quad (3)$$

The participant uses the belief distribution over states to compute the probability that the motion direction is leftwards ($k = -1$) and rightwards ($k = 1$):

$$\begin{aligned} P(k = -1|x) &= \frac{\sum_{m \in M} P(x|m, k = -1)}{\sum_{k'}\sum_{m \in M} P(x|m, k')} \\ P(k = 1|x) &= \frac{\sum_{m \in M} P(x|m, k = 1)}{\sum_{k'}\sum_{m \in M} P(x|m, k')} \end{aligned} \quad (4)$$

The participant makes their decision, $r$, by comparing the posteriors over motion direction:

$$r(x) = \begin{cases} -1 & \text{if } P(k = -1|x) > P(k = 1|x) \\ 1 & \text{if } P(k = -1|x) < P(k = 1|x) \end{cases} \quad (5)$$

Finally, the probability that the choice is correct can be computed directly using the posteriors over motion direction relative to the chosen action:

$$P(correct|x) = P(k = r|x) = \begin{cases} P(k = -1|x) & \text{if } r = -1 \\ P(k = 1|x) & \text{if } r = 1 \end{cases} \quad (6)$$

Other-trials: Participants were paired with three players of varying ability, $j \in J = \{1, 2, 3\}$. We consider below different models of how participants compute confidence in player $j$'s choices and how participants learn about player $j$'s ability.

Self-projection (S-model): A simple way to compute confidence in another player's choice is to apply the same model to the other player as one applies to oneself. Under this model, the participant simulates what choice they themselves would have made and how confident they would have felt about this choice – as if they were the other player.

The other player's posteriors over motion direction are calculated as:

$$\begin{aligned} P(k = -1|x, j) &= P(k = -1|x, s) \\ P(k = 1|x, j) &= P(k = 1|x, s) \end{aligned} \quad (7)$$

The other player's choice is calculated as:

$$r(x) = \begin{cases} -1 & \text{if } P(k = -1|x, j) > P(k = 1|x, j) \\ 1 & \text{if } P(k = -1|x, j) < P(k = 1|x, j) \end{cases} \quad (8)$$

The participant's confidence in the other player's choice is then calculated as:

$$P(correct|x, j) = P(k = r|x, j) = \begin{cases} P(k = -1|x, j) & \text{if } r = -1 \\ P(k = 1|x, j) & \text{if } r = -1 \end{cases} \quad (9)$$

Note that the S-model does not learn about the other player's ability.

Simple value learner (Q-model): An alternative strategy for estimating confidence in another player's choice is to keep a running estimate of the value (correctness) of their choices, $Q(j)$, where $j$ is the other player. This can be done compactly using a

Rescorla-Wagner update rule:

$$Q(j)_{t+1} := Q(j)_t + \alpha\delta_t \quad (10)$$

where $t$ indicates the trial, $\alpha$ is a learning rate and $\delta_t$ is the difference between the current outcome $o_t$ (correct $= 1$; incorrect $= 0$) and the current estimate, $Q(j)_t$:

$$\delta_t = o_t - Q(j)_t \quad (11)$$

The participant's confidence in the other player's choice is then calculated as:

$$P(correct|j) = Q(j) \quad (12)$$

We set $Q(j)_{t=0} = 0.75$ for each player $j$. Note that, unlike the S-model, the Q-model is blind to the effect of coherence on the other player's chance of success.

Theory of mind (ToM-model): A more sophisticated strategy is to maintain a running estimate of another player's sensory noise – that is, a model of their sensory system – and use this estimate to compute confidence in their choices. In the absence of shared error variance between the participant's and the other player's sensory sample (i.e., no noise correlation), the participant's prediction about the other player's success should take into account (1) the posterior probability that the world is in a given state, $(k, m)$, given their own sensory sample, $x$, and (2) the other player's state-dependent performance (i.e., their expected accuracy in each state) given their estimated sensory noise, $\sigma_j$. The prediction about the other player's success is thus given by:

$$P_j(correct|x) = \sum_k \sum_{m \in M} P(m, k|x)P_j(correct|m, k) \quad (13)$$

where $P_j$ denotes the probability that the other agent $j$ makes a correct response. The first term, the belief distribution over states, is calculated using $\sigma_s$, and the second term, the state-dependent performance of the other player, is calculated using $\sigma_j$:

$$P_j(correct|m, k) = \begin{cases} \phi(0; k\theta_m, \sigma_j) & \text{if } k\theta_m < 0 \\ 1 - \phi(0; k\theta_m, \sigma_j) & \text{if } k\theta_m > 0 \end{cases} \quad (14)$$

where $\phi$ is the cumulative normal distribution:

$$\phi(y; \mu, \sigma) = \frac{1}{\sigma\sqrt{2\pi}}\int_{-\infty}^{y} e^{\frac{-(z-\mu)^2}{2\sigma^2}} dz \quad (15)$$

The estimate of the other player's sensory noise is updated using an approximation to the Rescorla-Wagner rule:

$$\sigma_{j,t+1} = \sigma_{j,t} + \alpha\delta_t/D(x, \sigma_{j,t}) \quad (16)$$

where $\delta$ is a prediction error as in the Q-model and $D(x, \sigma_j)$ is the following derivative:

$$D(x, \sigma_j) = \frac{dP_j(correct|x)}{d\sigma_j} \quad (17)$$

In short, we want to define an update to the estimate of the other player's sensory noise, $\eta_t$, such that the prediction about the other player's success, $P_j(correct|x, \sigma_{j,t} + \eta_t)$, will be equal to $P_j(correct|x, \sigma_{j,t}) + \alpha\delta_t$:

$$P_j(correct|x, \sigma_{j,t} + \eta_t) = P_j(correct|x, \sigma_{j,t}) + \alpha\delta_t \quad (18)$$

We can approximate the left-hand term around $\sigma_{j,t}$ linearly. So, we obtain:

$$P_j(correct|x, \sigma_{j,t}) + D(x, \sigma_{j,t})\eta_t = P_j(correct|x, \sigma_{j,t}) + \alpha\delta_t \quad (19)$$

Or

$$\sigma_{j,t+1} = \sigma_{j,t} + \alpha\delta_t/D(x, \sigma_{j,t}) \quad (20)$$

The derivative of the prediction with respect to the parameter is calculated as:

$$\begin{aligned} D(x, \sigma_j) &= \sum_{m \in M, k = \pm 1}\left(\frac{dP(m, k|x)}{d\sigma_j}P_j[correct|m, k] + P(m, k|x)\frac{dP_j[correct|m, k]}{d\sigma_j}\right) \\ &= \sum_{m \in M, k = \pm 1}\left(P(m, k|x)\frac{dP_j[correct|m, k]}{d\sigma_j}\right) \end{aligned} \quad (21)$$

Because $P(m, k|x)$ is not a function of $\sigma_j$, its derivative is 0.

The term $\frac{dP_j[correct|m, k]}{d\sigma_j}$ is calculated as:

$$\frac{dP_j[correct|m, k]}{d\sigma_j} = -\frac{\theta_m}{\sigma_j^2\sqrt{2\pi}}e^{\frac{-(\theta_m)^2}{2\sigma_j^2}} = -\frac{\theta_m}{\sigma_j}\varphi(0; \theta_m, \sigma_j) \quad (22)$$

We thus have:

$$D(x, \sigma_j) = -\frac{1}{\sigma_j}\sum_{m \in M, k = \pm 1}[P(m, k|x)\theta_m\varphi(0; \theta_m, \sigma_j)] \quad (23)$$

The linear approximation to the Rescorla-Wagner rule may result in numerical overflow in a small number of cases (e.g., when $\sigma_j$ is extremely low and $x$ is extremely high). To avoid this issue, we implemented a proposal mechanism under which we

iteratively adjusted the update term, $\alpha\delta_t/D(x, \sigma_{j,t})$, until $|P_j(\text{correct}|x, \sigma_{j,t} + \eta_t) - (P_j(\text{correct}|x, \sigma_{j,t}) + \alpha\delta_t)| < 0.001$.

Post-decision wagering: In all models, PDWs are based on the expected values of the risky and the safe options:

$$EV_{risky} = P(\text{correct}|x, a)V_{risky} - [1 - P(\text{correct}|x, a)]V_{risky}$$
$$EV_{safe} = V_{safe} \qquad (24)$$

where $a$ indicates the agent (participant or other) and $V$ indicates the option value.

To account for individual differences in the propensity to gamble on a decision and sensitivity to changes in expected value, we modelled the probability of selecting the risky option using a softmax function:

$$P(risky) = \frac{1}{1 + e^{-[\beta_0 + \beta_1(EV_{risky} - EV_{safe})]}} \qquad (25)$$

Model variations: We considered versions of the above models where the softmax bias, $\beta_0$, and/or temperature, $\beta_1$, could vary between self- and other-trials and where the representation of the stimulus space, $M$, was rescaled according to a power law (using a factor $p$). In brief, we rescaled each state as $m' = m^p$ and normalised the resulting state space, $M'$, using the minimum and maximum values of the original stimulus space, $M$. The rescaling allowed for a higher density of motion stimuli at low ($p > 1$) or high ($p < 1$) stimulus intensities (where $p = 1$ preserves the original stimulus space). There were thus 24 models: 3 (S, Q, ToM) × 2 (same versus different bias, $\beta_0$, for self and other) × 2 (same versus different temperature, $\beta_1$, for self and other) × 2 (linearly-spaced versus rescaled stimulus space).

*Model fitting and comparison.* We used variational Bayesian inference as implemented in Stan[58] to fit the models to the behavioural data. We used a hierarchical fitting procedure where group-level parameters constrain participant-level parameters (see Supplementary Table 1 for full parameter space). We fitted each model using the following specifications: maximum number of iterations = 4000; number of samples for Monte Carlo estimation of objective function = 200; number of iterations between evaluation of objective function = 100; convergence tolerance on the relative norm of the objective = 0.0001. To generate trial-by-trial predictions about each participant's behaviour, and obtain estimates of the latent model variables, we drew 500 samples from the posterior distributions over fitted parameters (the 'generated quantities' block in Stan), calculated the trial-by-trial values for each set of samples and then averaged across these 500 samples. To assess the ability of each model to account for a participant's data, we computed the log-likelihood of their trial-by-trial PDWs under each draw of parameters and applied leave-one-out cross-validation using Pareto-smoothed importance sampling (PSIS-LOO) and the widely applicable information criterion (WAIC) – both methods for estimating pointwise out-of-sample prediction accuracy under a Bayesian model[59]. We performed these steps 4 times for each model using different random seeds and averaged all outputs across the 4 runs. The results of the model comparison are shown in Supplementary Fig. 2.

*Model identifiability.* As a quality control of our modelling approach, we conducted a model identifiability analysis[29]. We selected the best-fitting version of each model class (Supplementary Fig. 2), simulated 200 datasets for each of the three models and then fitted the three models to the simulated datasets; each dataset mirrored our study in terms of participants, task variables and trials. Based on the model comparison results for all 200 iterations, we computed a confusion and an inversion matrix. A confusion matrix shows the probability that model Y provided the best fit to data generated by model X. By contrast, an inversion matrix shows the probability that model X generated the data given that model Y provided the best fit – in other words, if model Y wins the model comparison, which model is most likely to be the true model? As shown in Supplementary Fig. 5, the model identifiability analysis shows that all three models were discriminable within the constraints of our experimental paradigm.

*Risk and/or loss aversion.* Our models inferred participants' confidence based on PDWs. Previous research has shown that PDWs are subject to risk and loss aversion[30] – raising the possibility that these factors may have biased the model comparison and/or distorted the model-based estimates of confidence for fMRI analysis. To rule out this possibility, we selected the best-fitting version of each model class (Supplementary Fig. 2) and re-fitted these models to the behavioural data after including a utility function. This function transforms the expected values of the gamble options into subjective utilities and allows for individual variation in risk and/or loss aversion. Formally, we computed the expected values of the risky and the safe options as follows:

$$EV_{risky} = P(\text{correct}|x, a)V_{risky}^R - [1 - P(\text{correct}|x, a)]SV_{risky}^T$$
$$EV_{safe} = V_{risky}^R \qquad (26)$$

where $R$, $T$ and $S$ parameterise the utility function, with $S$ controlling loss aversion and $R$ and $T$ controlling risk-seeking/risk-aversion in the gain and loss domains,

respectively. In keeping with previous work[30], we restricted $R$ and $T$ to the range 0–1 (see Table S1 for all parameters). The ToM-model continued to provide the best fit to the behavioural data, and the model-based confidence estimates were highly correlated (Pearson's $r > 0.9$ for all participants) between the original ToM-model and the ToM-model with a utility function (Supplementary Fig. 6).

For completeness, we visualised the fitted group-level utility function under the ToM-model. In line with classic behavioural economics results[60] and earlier work on the drivers of PDWs[30], this visualisation showed that losses loomed larger than gains (Supplementary Fig. 6).

Although not shown, we highlight that all model-based fMRI results remained the same when performed using the ToM-model with a utility function.

*FMRI*

Whole-brain general linear models: The whole-brain analysis shown in Fig. 4 was based on a single event-related general linear model (GLM1). We separated trials into self-trials and other-trials and specified separate 'condition' regressors for the decision phase (i.e., boxcar from RDK onset to 3 s after decision onset) and the gamble phase (i.e., boxcar from gamble onset until feedback offset) for each trial type (yielding 4 condition regressors in total). We included motion parameters as 'nuisance' regressors. Regressors were convolved with a canonical hemodynamic response function. Regressors were modelled separately for each scan run and constants were included to account for between-run differences in mean activation and scanner drifts. A high-pass filter (128 s cut-off) was applied to remove low-frequency drifts. Group-level significance was assessed by applying one-sample $t$-tests against zero to the first-level contrast images. We report clusters significant at $p < 0.05$, FWE-corrected for multiple comparisons, with a cluster extent of 10 voxels or more and a cluster-defining threshold of $p < 0.001$, uncorrected. Numerical simulations and tests of empirical data collected under the null hypothesis show that this combination of cluster-defining threshold and random field theory produces appropriate control of false positives[61].

Regions of interest: ROI masks for MT+ were created using the localiser scan: we created a group mask using the second-level contrast between dynamic and static motion, and then, for each participant, created a MT+ mask (8-mm sphere) around their peak activity inside the group mask. ROI masks for LIP, TPJ and dmPFC were created using published connectivity-based parcellation atlases: LIP was defined as the union of areas SPLD and SPLE in the atlas developed by Mars et al.[31]; TPJ was defined as area TPJp in the atlas developed by Mars et al.[62]; and dmPFC was defined as area 9 in the atlas developed by Neubert et al.[63]. All ROI masks were bilateral.

Psychophysiological interaction analysis: To assess changes in connectivity between visual and social ROIs as a function of trial type, we carried out a psychophysiological interaction (PPI) analysis using the Generalised PPI toolbox for SPM (gPPI; http://www.nitrc.org/projects/gppi/). The toolbox takes one GLM as its input (here, GLM1) and then creates a new GLM in which the deconvolved activity of the seed region (MT+ or LIP) is assigned to separate regressors dependent on the status of the psychological variable (self versus other) and re-convolved with a canonical hemodynamic response function. We extracted average contrast estimates within an ROI for each participant and then used these estimates for group-level testing.

Single-trial activity time courses: We transformed each ROI mask from MNI to native space and extracted preprocessed BOLD time courses as the average of voxels within the mask. For each scan run, we regressed out variation due to head motion as specified above, applied a high-pass filter (128 s cut-off) to remove low-frequency drifts, and oversampled the BOLD time course by a factor of ~23 (time resolution of 0.144 s; spline interpolation). For each trial, we extracted activity estimates in a 14 s window around an event of interest (RDK or feedback onset).

Canonical hemodynamic response functions: In addition to the whole-brain GLM analyses described above, we estimated whole-brain GLMs which included a separate regressor for each trial[64]. For the stimulus-related analyses, the regressors were boxcars spanning RDK presentation. For the feedback-related analysis, the regressors were boxcars spanning feedback delivery. Each of these regressors was convolved with a canonical hemodynamic response function. We included motion parameters as 'nuisance' regressors. Regressors were modelled separately for each scan run and constants were included to account for between-run differences in mean activation and scanner drifts. A high-pass filter (128 s cut-off) was applied to remove low-frequency drifts.

One consideration when obtaining single-trial activity estimates as a beta time series is that a beta for a given trial can be affected by acquisition artefacts that occur together with that trial (e.g., scanner pulse artefacts). Therefore, for each participant, we computed the grand-mean beta estimate across both voxels and trials and excluded any trial whose mean beta estimate across voxels was 3 SDs below or above this grand mean[64]. Finally, we used the ROI masks to extract single-trial ROI activity estimates under the canonical hemodynamic response function.

**Software**. The task was programmed in MATLAB 2014b using Psychtoolbox 3.0.12[65,66]. The behavioural data were analysed in MATLAB 2015b. Neural data

were analysed in MATLAB 2015b using SPM12 and the Generalised PPI toolbox 13.1. Computational models were fitted using RStudio 1.0.153 and Stan 2.19.1.

**Reporting summary**. Further information on research design is available in the Nature Research Reporting Summary linked to this article.

## Data availability

Data supporting behavioural and neural analyses are available on GitHub: https://github.com/danbang/article-self-other. Unthresholded group-level statistical maps are available on NeuroVault: https://neurovault.org/collections/9553/. Source data are provided with this paper.

## Code availability

Code for reproducing behavioural and neural analyses are available on GitHub: https://github.com/danbang/article-self-other.

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

## Acknowledgements

We thank Sara Ershadmanesh and Matthew Rushworth for helpful comments on an earlier version of the manuscript. The Wellcome Centre for Human Neuroimaging is supported by core funding from the Wellcome Trust (203147/Z/16/Z). The Max Planck UCL Centre for Computational Psychiatry and Ageing Research is funded by a joint initiative between UCL and the Max Planck Society. D.B. was supported by a Sir Henry Wellcome Postdoctoral Fellowship funded by the Wellcome Trust (213630/Z/18/Z). N.D.D. was supported by the John Templeton Foundation (61454). S.M.F. was supported by a Sir Henry Dale Fellowship jointly funded by the Wellcome Trust and the Royal Society (206648/Z/17/Z). For the purpose of Open Access, the authors have applied a CC BY public copyright licence to any Author Accepted Manuscript version arising from this submission.

## Author contributions

S.M.F. and N.D.D. conceived the study. S.M.F. collected the data. D.B., R.M., N.D.D and S.M.F developed the models. D.B. analysed the data. D.B., R.M., N.D.D. and S.M.F. interpreted the results. D.B. wrote the manuscript. R.M., N.D.D. and S.M.F. provided critical comments. All authors approved the final manuscript.

## Competing interests

The authors declare no competing interests.
