## [Peer Review File · Nature Communications]

Neurocomputational mechanisms of confidence in self and othersReviewers' comments:

Reviewer #1 (Remarks to the Author):

Dan Bang and colleagues present results from a fMRI study (N=23) aimed at elucidating the neural correlates of “confidence” for other decisions. The administered a perceptual discrimination task followed by a wager decision (i.e., whether or not to bet on the correctness of a perceptual choice). The key manipulation of the study is a ‘self/other’ contrast, where in half of the trials the choice is made by the participant (self) and in the other half of the trial is made by other subjects previously recorded. Crucially the other players displayed three levels of ‘ability’ (perceptual accuracy). The authors analyse the $p(\text{gamble})$ (as a proxy of confidence) and found that (among other factors) it is affected by the skills of the other subject. The authors propose a model where the subjects progressively infer the ability of the other subject to modulate gamble decisions in the ‘other’ trials. The paper reads quite well (except for some of fMRI parts) and the question is interesting. However, I do not believe that the paper presents compelling evidence concerning the proposed computational model.

1/ The first major issue concerns the fact that the current analysis infers the ability of the other subjects based on gamble choices. Unfortunately this inference is valid only assuming that gambling choices are solely due to the perceived task difficulty and the inferred ability: an assumption that can hardly be held in the current paradigm, where $p(\text{gamble})$ will also be affected by non-social and non-perceptual parameters, such as risk and loss aversion. Without taking into account these factors, the authors are conflating the true confidence about the other’s accuracy with completely private and value-based processes. For a full treatment (the authors should be aware of...) of the issues related to inferring confidence from wager tasks see:

<https://www.ncbi.nlm.nih.gov/pmc/articles/PMC2842936/>

2/ Subjects’ behavior and model properties are not fully explored. All models are learning models, yet time series of the key latent and predicted variables are missing. For instance what is the temporal dynamics of $p(\text{gamble})$ in the different ability and reward treatments in the ‘other’ conditions? Do the models capture the temporal evolution of this variable? Also, related to my first point, what is predicted (on average and over time) about $p(\text{correct})$ in the same conditions?

3/ Looking at the model simulations in the supplementary material, all models (except the Q*-model - that I consider a strawman because there are no explicit categories corresponding to the difficulty levels in the task that the subject should/could learn independently) predict very similar behavioral patterns it is therefore unclear what feature of the data is favouring the selected model - see Palminteri et al. 2017. A related point is that the authors should provide model identifiability / recovery and confusion matrix to ensure that the models are discriminable within their behavioral paradigm (conditions, number of trials) - see Wilson and Collins 2019. I suspect that this will not be the case because of the relatively low number of trials.

4/ This is rather a conceptual / interpretational point, but with implications concerning the interpretation of the fMRI results. The tasks involved no actual social interaction (cooperation or competition) and everything needed to infer the others’ ability is overt behavior (past correct response, and objective difficulty), so why would this task involve “theory of mind” at all? What is

the others' mental state that is inferred here? The exact same task (predicting $p(\text{correct})$ for a pattern detection task for different detector performance) could be framed - and I bet solved by participants - in a completely social-free framing. Furthermore, the participants are told that there the other players have been recorded, making mental state inference even less useful. This lack of explicit need for ToM in the task (IMHO) also challenges the fMRI "predictions" outlined by the others.

5/ Concerning the fMRI results I will make more generic comments on the first results, as other results (e.g., functional connectivity) rely on the validity of the computational model, which I questioned in the points above). My first remark is that the sample size and the trial numbers are both below the standards of the field.

6/ The activations of the SELF>Other contrast presents a big confound in that the self condition demands motor preparation and execution. This extra decision and motor execution related to the selection of the chosen option could actually explain most of the neural network revealed in Fig 4 top, which seems to include the preSMA, (wrongly labeled as dACC) as well as (left) primary motor cortex (labelled as posterior parietal). This challenges the current authors' interpretations of those activations as "perceptual". Also, I do not get the rationale explaining why classic "perceptual decision-making regions – including MT+ and LIP – [should be] more active on self- than other-trials during the decision phase" (lines 306-307)

Reviewer #3 (Remarks to the Author):

Bang et al studied how confidence in the decision of another person is computed. In a task that manipulated the proficiency of the other "person" at the task and the difficulty of the decision itself, this study shows that observers take both discrimination difficulty and the proficiency of the person being observed into account when computing confidence in others' choices. Concurrently recorded BOLD signals show that TPJ and dmPFC signals encoded observers trial-by-trial estimates of confidence in others and 'social prediction' errors, suggesting these areas might be involved in the computation of confidence in others. This is an interesting and timely paper using a novel experimental design. I have a few comments that the authors should be able to address.

1. The models used to fit the behavioral data and explain fMRI responses are all grounded in Bayesian decision theory. However, Bayesian models of confidence have not received strong empirical support as there are many examples of papers showing that confidence ratings do not track the probability of being correct (Koizumi et al., 2015; Maniscalco et al., 2016; Samaha et al., 2016; Zylberberg et al., 2012, 2014) and that Bayesian models do not necessarily fit confidence data the best (Adler & Ma, 2018; Denison et al., 2018; Li & Ma, 2020). To the extent that the model-predictions are used in regression analysis to explain fMRI responses, it would seem that basing the model on incorrect assumptions about how confidence is computed could impact the nature and interpretation of the fMRI results.

2. I found it a bit difficult to understand how specific some effects were as comparison between tasks were not always reported. For instance, for the analysis presented in figure 7 – my understanding is that only 'other' trials were used to predict bold responses in TPJ and dmPFC,

which showed encoding of model-based estimates of others' confidence in these regions, but is this response specific to that task or do these regions also show encoding of confidence in general (i.e., in oneself). Since different regions of PFC have been suggested to perform metacognitive functions (Rahnev et al., 2016) it's unclear if the authors think that dmPFC is specific for evaluating confidence in others. They could report whether the relationship between other's confidence and TPJ/dmPFC activity is stronger than the relationship between one's own confidence and TPJ/dmPFC activity.

3. I am also a bit confused about the interpretation of the LIP area encoding of coherence in self-versus other trials (e.g., figure 6). It's quite striking that LIP encoding of coherence is reversed for self-versus other trials – importantly, this would seem to go against the idea that the same evidence is used for self and other confidence computations. That is, I thought the authors argue that a key variable in the ToM account is a representation of task difficulty akin to computing confidence in one's own choice, but the fact that LIP represents motion evidence so differently in self versus other tasks would seem, at least at first blush, inconsistent with this notion.

4. lastly, was any kind of correction for multiple comparisons used in the time-course analyses? It seems like some differences are present even in the pre-stimulus window (e.g., figure 6)

Sincerely,
Jason Samaha

Adler, W. T., & Ma, W. J. (2018). Comparing Bayesian and non-Bayesian accounts of human confidence reports. *PLOS Computational Biology*, 14(11), e1006572.

<https://doi.org/10.1371/journal.pcbi.1006572>

Denison, R. N., Adler, W. T., Carrasco, M., & Ma, W. J. (2018). Humans incorporate attention-dependent uncertainty into perceptual decisions and confidence. *Proceedings of the National Academy of Sciences*, 115(43), 11090–11095. <https://doi.org/10.1073/pnas.1717720115>

Koizumi, A., Maniscalco, B., & Lau, H. (2015). Does perceptual confidence facilitate cognitive control? *Attention, Perception & Psychophysics*, 77(4), 1295–1306.

<https://doi.org/10.3758/s13414-015-0843-3>

Li, H.-H., & Ma, W. J. (2020). Confidence reports in decision-making with multiple alternatives violate the Bayesian confidence hypothesis. *Nature Communications*, 11(1), 1–11.

<https://doi.org/10.1038/s41467-020-15581-6>

Maniscalco, B., Peters, M. A. K., & Lau, H. (2016). Heuristic use of perceptual evidence leads to dissociation between performance and metacognitive sensitivity. *Attention, Perception, & Psychophysics*, 1–15. <https://doi.org/10.3758/s13414-016-1059-x>

Rahnev, D., Nee, D. E., Riddle, J., Larson, A. S., & D'Esposito, M. (2016). Causal evidence for frontal cortex organization for perceptual decision making. *Proceedings of the National Academy of Sciences*, 113(21), 6059–6064. <https://doi.org/10.1073/pnas.1522551113>

Samaha, J., Barrett, J. J., Sheldon, A. D., LaRocque, J. J., & Postle, B. R. (2016). Dissociating Perceptual Confidence from Discrimination Accuracy Reveals No Influence of Metacognitive Awareness on Working Memory. *Frontiers in Psychology*, 7.

<https://doi.org/10.3389/fpsyg.2016.00851>

Zylberberg, A., Barttfeld, P., & Sigman, M. (2012). The construction of confidence in a perceptual decision. *Frontiers in Integrative Neuroscience*, 6.
<https://doi.org/10.3389/fnint.2012.00079>

Zylberberg, A., Roelfsema, P. R., & Sigman, M. (2014). Variance misperception explains illusions of confidence in simple perceptual decisions. *Consciousness and Cognition*, 27, 246–253. <https://doi.org/10.1016/j.concog.2014.05.012>

REVIEWER 1

Dan Bang and colleagues present results from a fMRI study (N=23) aimed at elucidating the neural correlates of “confidence” for other decisions. They administered a perceptual discrimination task followed by a wager decision (i.e., whether or not to bet on the correctness of a perceptual choice). The key manipulation of the study is a ‘self/other’ contrast, where in half of the trials the choice is made by the participant (self) and in the other half of the trial is made by other subjects previously recorded. Crucially the other players displayed three levels of ‘ability’ (perceptual accuracy). The authors analyse the p(gamble) (as a proxy of confidence) and found that (among other factors) it is affected by the skills of the other subject. The authors propose a model where the subjects progressively infer the ability of the other subject to modulate gamble decisions in the ‘other’ trials. The paper reads quite well (except for some of fMRI parts) and the question is interesting. However, I do not believe that the paper presents compelling evidence concerning the proposed computational model.

We thank the reviewer for the positive assessment of our manuscript, and for their detailed suggestions for how to test the robustness of our modelling approach. We have now implemented these suggestions and hope that the new results detailed below address their remaining concerns.

(1.1) The first major issue concerns the fact that the current analysis infers the ability of the other subjects based on gamble choices. Unfortunately, this inference is valid only assuming that gambling choices are solely due to the perceived task difficulty and the inferred ability: an assumption that can hardly be held in the current paradigm, where p(gamble) will also be affected by non-social and non-perceptual parameters, such as risk and loss aversion. Without taking into account these factors, the authors are conflating the true confidence about the other’s accuracy with completely private and value-based processes. For a full treatment (the authors should be aware of...) of the issues related to inferring confidence from wager tasks see: <https://www.ncbi.nlm.nih.gov/pmc/articles/PMC2842936/>

We thank the reviewer for this suggestion. We have now implemented a control analysis in which we test whether the inclusion of *non-social and non-perceptual value-based processes* changes our results. To this end, we selected the best-fitting version of each model class (self-projection, S; performance-tracking, Q; theory of mind, ToM; see **point (1.3)** for why we now omit Q* and ToM*) and re-fitted these models after including a utility function – using the formalisation in the cited paper (Fleming & Dolan, *Consciousness & Cognition*, 2010). The utility function transforms the expected values of the gamble options into subjective utilities and allows for individual differences in risk and/or loss aversion. Formally, we calculated the expected values of the risky and the safe options as follows:

$$EV_{risky} = confidence \times V_{risky}^R - (1 - confidence) \times S \times V_{risky}^T$$
$$EV_{safe} = V_{safe}^R$$

Where V is the option value and R , T and S parameterise the utility function – with S controlling loss aversion and R and T controlling risk-seeking/risk-aversion in the gain and loss domains, respectively. As described below, we find that the conclusions in the original manuscript remain unchanged.

First, our winning model, the ToM-model, still provides the best account of subjects’ behaviour:

Second, the model-based confidence estimates under the original ToM-model and the ToM-model with a utility function are almost perfectly correlated. Here we show the Pearson's correlation coefficient, r , between the model-based confidence estimates under the two models for each subject and trial type:

This near-perfect correspondence can also be seen when overlaying their behavioural predictions (the dark shaded band indicates the mean prediction \pm 95% confidence intervals under the original ToM-model; the pink line indicates the mean prediction under the ToM-model with a utility function):

The above plots are based on “**Figure 2: Behavioural results**” in the original manuscript.

Finally, the original model-based fMRI results – where we quantified neural encoding of model-based confidence estimates orthogonalised with respect to motion coherence (left column) and model-based prediction errors (right column) – remain the same whether based on the original ToM-model fit (top row) or on the fit of the ToM-model with a utility function (bottom row):

The above plots are based on “**Figure 7: TPJ and dmPFC support a social confidence computation**” and “**Figure 9: Encoding of social prediction errors in TPJ and dmPFC**” in the original manuscript.

For completeness, we also visualised the inferred group-level utility function under the ToM-model:

In keeping with classic results in behavioural economics and our own earlier results on the drivers of post-decision wagers (Fleming & Dolan, *Consciousness & Cognition*, 2010), this visualisation showed that losses loomed larger than gains.

In summary, while we agree that non-social and non-perceptual value-based processes are at play, they do not preclude inferring confidence from gamble choices, and controlling for their influence does not affect our original results. The option values were, by design, randomised across conditions and it is therefore unlikely that risk and/or loss aversion would have driven the observed variation in subjects’ gambling behaviour as a function of decision difficulty and/or others’ ability. Because the simpler models without a utility function produce similar results, and because the task design controls for value-based processes, we continue to rely on the simpler models in the revised manuscript, and we instead include the above analysis as a control analysis in the Supplementary Information (new Figure S6).

We have made the following changes to the manuscript:

(1.1.A) When describing the task, we now say [lines 110-111]:

We highlight that our three main factors – motion coherence, others' ability and the reward difference between the risky and the safe option – were, by design, uncorrelated.

(1.1.B) In the modelling section, we now say [lines 256-264]:

[...] the models infer subjects' true confidence from their PDWs – raising the possibility that risk and/or loss aversion may have biased the model comparison results and/or distort the model-based estimates of confidence for subsequent fMRI analysis³¹. To rule out this possibility, we re-fitted the best-fitting version of each model class after adding a utility function, which transforms the expected value of the gamble options into subjective utilities, and allow for individual differences in risk and/or loss aversion³¹. This analysis confirmed the ToM-model as providing the best account of the behavioural data and showed that the model-based estimates of confidence remained the same when inferred with or without a utility function (**Figure S6**).

(1.1.E) We have included a description of the control analysis in the Methods [lines 796-815]:

Our models inferred subjects' confidence based on PDWs. Previous research has shown that PDWs are subject to risk and loss aversion¹ – raising the possibility that these factors may have biased the model comparison results and/or distorted model-based estimates of confidence used in fMRI analysis. To rule out this possibility, we selected the best-fitting version of each model class (**Figure S2**) and re-fitted these models to the behavioural data after including a utility function. This function transforms the expected values of the gamble options into subjective utilities and allows for individual variation in risk and/or loss aversion. Formally, we computed the expected values of the risky and the safe options as follows:

$$EV_{risky} = P(\text{correct}|x, a)V_{risky}^R - [1 - P(\text{correct}|x, a)]SV_{risky}^T$$
$$EV_{safe} = V_{risky}^R$$

Eq. 41

where R , T and S parameterise the utility function – with S controlling loss aversion and R and T controlling risk-seeking/risk-aversion in the gain and loss domains, respectively. In keeping with previous work³¹, we restricted R and T to the range 0-1 (see **Table S1** for all parameters). Importantly, the ToM-model continued to provide the best fit to the behavioural data, and the model-based confidence estimates were highly correlated (Pearson's $r > 0.9$ for all subjects) between the original ToM-model and the ToM-model with a utility function (**Figure S6**).

For completeness, we visualised the fitted group-level utility function under the ToM-model. In keeping with classic results in behavioural economics⁶⁶ and earlier results on the drivers of PDWs³¹, this visualisation showed that losses loomed larger than gains (**Figure S6**).

Although not shown here, we highlight that all model-based fMRI results remained the same when performed using the ToM-model with a utility function.

(1.1.D) We have created a supplementary figure to report this control analysis:

Figure S6. Risk and/or loss aversion. To assess the potential impact of risk and/or loss aversion on the model-based confidence estimates, we ran a control analysis in which we re-fitted the best-fitting version of each model class (S-B2T2P1, Q-B2T2P1 and ToM-B2T2P1) after including a utility function. **(A)** The ToM-model still provided the best account of the behavioural data (sum of WAIC across subjects) – indicating that value-based processes did not bias our model comparison results. **(B)** The model-based confidence estimates under both the original ToM-model and the ToM-model with a utility function were highly correlated (Pearson’s r) within every subject – supporting that the original ToM-model was sufficient to infer subjects’ confidence from their PDWs. **(C)** Visualisation of the inferred group-level utility function obtained from the fit of the augmented ToM-model. We normalised the value range for ease of visualisation.

(1.2) Subjects’ behavior and model properties are not fully explored. All models are learning models, yet time series of the key latent and predicted variables are missing. For instance what is the temporal dynamics of $p(\text{gamble})$ in the different ability and reward treatments in the ‘other’ conditions? Do the models capture the temporal evolution of this variable? Also, related to my first point, what is predicted (on average and over time) about $p(\text{correct})$ in the same conditions?

We thank the reviewer for this suggestion. We originally only showed data averaged across time, except for “**Figure 7: TPJ and dmPFC support a social confidence computation**” which showed a single-subject example of the trial-by-trial evolution of the estimates of others’ ability (sensory noise) and the confidence in others’ choices as computed under the winning ToM-model. We agree with the reviewer that a more complete picture is achieved by visualising the temporal dynamics of both behavioural data and model predictions at the group level. We have now performed these visualisations which show that the winning ToM-model captures the temporal evolution of $P(\text{gamble})$ in the behavioural data (as shown by an overlap of 95% confidence intervals) and provide an intuition for how the latent model parameters change with task experience. We note that we can only group the temporal dynamics by others’ ability as decision difficulty and option values were randomised across trials within a block. We highlight that we fit our models at the trial-by-trial level and that the winning candidate model is therefore the model which best explains the empirically observed temporal evolution of $P(\text{gamble})$.

We have made the following changes to the manuscript:

(1.2.A) In the modelling section, we now say [lines 248-251]:

As shown in **Figure 2**, the ToM-model captured all qualitative features of the behavioural data (see **Figure S3** for behavioural predictions under the best-fitting version of each model class). In addition, the ToM-model captured the trial-by-trial evolution of subjects' PDWs for each of the three other players (**Figure S4**).

(1.2.B) We have created a supplementary figure to support this conclusion:

Figure S4. Temporal dynamics of model variables. **(A)** Evolution of probability gamble within each block as observed in the behavioural data and as predicted under the ToM-model. **(B)** Evolution of the estimate of the other players' sensory noise under the ToM-model. **(C)** Evolution of confidence in the other players' choice under the ToM-model. For each variable, we applied a running average where the value on trial t is the average over trials $t-2$ to t – the data are therefore shown from trial 3 onwards. Empirical and simulated data are represented as group mean \pm 95% CI.

(1.3) Looking at the model simulations in the supplementary material, all models (except the Q*-model – that I consider a strawman because there are no explicit categories corresponding to the difficulty levels in the task that the subject should/could learn independently) predict very similar behavioural patterns it is therefore unclear what feature of the data is favouring the selected model – see Palminteri et al. 2017. A related point is that the authors should provide model identifiability / recovery and confusion matrix to ensure that the models are discriminable within their behavioral paradigm (conditions, number of trials) – see Wilson and Collins 2019. I suspect that this will not be the case because of the relatively low number of trials.

We thank the reviewer for this suggestion which prompted us to refine our modelling approach. Our three model classes in fact make qualitatively different predictions – a strength of our study which we regret that we did not clarify – and model identifiability analysis as suggested by the reviewer shows that the three model classes are discriminable within the constraints of our behavioural paradigm. In addition, as we describe below, we have reduced the model set from five to three models – keeping the three main models but removing the two sub-models, which, as alluded to by the reviewer, are not fully suited for our behavioural paradigm.

Initially, our three main models were: the self-projection (S) model, the performance-tracking (Q) model and the theory of mind (ToM) model. We later included two sub-models based on recent developments in computational modelling of (non-social) confidence. First, the Q*-model learned the value of the different levels of motion coherence, building on animal models of confidence in the random dot motion task (Lak et al., *Current Biology*, 2017; Lak et al., *Neuron*, 2020). Second, the ToM*-model allowed for shared error variance between subjects' and the other players' perception, building on recent Bayesian frameworks for model-based confidence formation (Fleming & Daw, *Psychological Review*, 2017).

In hindsight, the Q*-model and the ToM*-model did not differ from the parent models in terms of their qualitative predictions, and more subtle quantitative differences were unlikely to be captured by our behavioural paradigm (a conclusion also supported by our model identifiability analysis, outlined below). More specifically, we agree with the reviewer that the Q*-model is better suited for a random dot motion task with a more discrete space of motion stimuli (e.g., three difficulty levels as in Lak et al., *Current Biology*, 2017). In addition, for successful inference on the parameter governing shared error variance, the ToM*-model ideally requires a task design in which subjects and the other player make a choice on every trial and subjects are informed about which choice the other player made. We have therefore decided to remove these two models (Q* and ToM*) from the revised manuscript. Importantly, this step does not change our conclusions, but serves to clarify our modelling section. For transparency, we continue to refer to all five models in our reply below, while noting that the key distinction we make in the revised manuscript is between the three qualitative model classes (S, Q, ToM), rather than within-class comparisons.

As mentioned above, we had designed the task such that the three model classes predict qualitatively different behavioural patterns, as recommended by Palminteri et al. (*Trends in Cognitive Science*, 2017). In other words, we are able to say exactly which feature of the data is favouring a particular model. We regret that we did not clearly explain this point. As shown below – and highlighting that parent models and sub-models are qualitatively similar – the ToM-models are the only model class which predict all qualitative features of subjects' confidence in others' decisions:

Model class		Predicts behaviourally observed impact of task variable on confidence in others' choices			
		Motion coherence	Reward difference	Others' ability	Previous choice accuracy
Self-projection	S	+	+	×	×
	Q	×	+	+	+
Performance-tracking	Q*	×	+	+	+
	ToM	+	+	+	+
Theory of mind	ToM*	+	+	+	+

We agree with the reviewer that model identifiability analysis is useful to demonstrate the robustness of our modelling approach. Following Wilson & Collins (*eLife*, 2019), we have now simulated data under each model and fitted each model to these data. We repeated this step 200 times, using the same number of subjects and trials and the same task variables as in our experiment. Based on these results, we computed confusion and inversion matrices. A confusion matrix shows the probability that model Y provided the best fit to data generated by model X. By contrast, an inversion matrix shows the probability that model X generated the data given that model Y provided the best fit – in other words, if model Y wins the model comparison, which model is most likely to be the true model? As Wilson & Collins highlight, the inversion matrix captures the goal of computational modelling in cognitive neuroscience – where we want to know which model among a set of candidate models most accurately describes the

underlying cognitive process. The figure below shows the confusion and inversion matrices (top: main + sub-models; bottom: after removing sub-models as done in the revised manuscript):

As highlighted above, there is considerable confusion between the Q- and Q*-models and between the ToM- and ToM*-models in the upper row. However, as can be seen from the warmer colours falling along the diagonal in the lower row, the three qualitatively distinct model classes are identifiable within the constraints of our behavioural paradigm. More importantly, the inversion matrix shows that, if a ToM-model provides the best fit to the data, then we can have a high degree of confidence that a ToM-like processes generated the data.

In summary, we now only focus on clearly identifiable model distinctions (S versus Q versus ToM). We believe that the *qualitative* and *quantitative* model diagnostics described above support the robustness of our modelling approach – and we thank the reviewer for prompting these analyses.

We have made the following changes to the manuscript:

(1.3.A) We have removed the Q*-model and the ToM*-model from the manuscript.

(1.3.B) We have revised the supplementary figure which shows behavioural predictions under the best-fitting version of each model class to better visualise the qualitative model distinctions and to include an overview of these differences:

Figure S3. Model predictions. In each panel, data are split into self-trials (blue) and other-trials (yellow). **(A)** Probability gamble as a function of coherence (median split). **(B)** Probability gamble as a function of the reward difference between the risky and the safe option (median split). **(C)** Probability gamble as a function of the ability of the other player. **(D)** Probability gamble as a function of choice accuracy on the previous trial of the same type. **(E)** While all models predict the observed pattern of results on self-trials, only the ToM-model predicts the observed pattern of results on other-trials. **(A-D)** Bar charts are empirical data. Dots show data simulated under the best-fitting version of each model class (S-B2T2P1, Q-B2T2P1 and ToM-B2T2P1). Models are denoted by colour and displayed in the following order: S-model, Q-model and ToM-model. Empirical data are represented as group mean \pm 95% CI and model predictions are represented as group mean.

(1.3.C) In the modelling section, we now say [lines 253-256]:

[...] we performed a model identifiability analysis using the best-fitting versions of each model class³⁰. Specifically, we simulated data under each model and then fitted each model to these data. This analysis showed that the models were discriminable within the constraints of our experimental paradigm **(Figure S5)**.

(1.3.D) We have created a supplementary figure to support this conclusion:

Figure S5. Model recoverability analysis. We selected the best-fitting version of each model class (S-B2T2P1, Q-B2T2P1 and ToM-B2T2P1), simulated 200 datasets for each model – with each dataset mirroring our study in terms of subjects, task variables and trials – and then fitted the models to the simulated datasets. Based on the model comparison results (WAIC) for all 200 iterations, we computed a confusion and an inversion matrix. A confusion matrix shows the probability that model Y provided the best fit to data generated by model X. By contrast, an inversion matrix shows the probability that model X generated the data given that model Y provided the best fit. We calculated the inversion matrix directly from the confusion matrix and assumed that the three models were a priori equally likely. Model classes are abbreviated as follows: S, S-model; Q, Q-model; T, ToM-model.

(1.3.E) We have included a description of the control analysis in the Methods [lines 783-795]:

As a quality control of our modelling approach, we conducted a model identifiability analysis³⁰. We selected the best-fitting version of each model class (**Figure S2**), simulated 200 datasets for each of the three models – with each dataset mirroring our study in terms of subjects, task variables and trials – and then fitted the three models to the simulated datasets. Based on the model comparison results for all 200 iterations, we computed a confusion and an inversion matrix. A confusion matrix shows the probability that model Y provided the best fit to data generated by model X. By contrast, an inversion matrix shows the probability that model X generated the data given that model Y provided the best fit – in other words, if model Y wins the model comparison, which model is most likely to be the true model? The inversion matrix is closely related to the goal of computational modelling in cognitive neuroscience – where we want to know which model among a set of candidate models best describes the underlying cognitive process. As shown in **Figure S5**, the model identifiability analysis shows that all three models were discriminable within the constraints of our experimental paradigm.

(1.4) This is rather a conceptual / interpretational point, but with implications concerning the interpretation of the fMRI results. The tasks involved no actual social interaction (cooperation or competition) and everything needed to infer the others' ability is overt behavior (past correct response, and objective difficulty), so why would this task involve "theory of mind" at all? What is the others' mental state that is inferred here? The exact same task (predicting p(correct) for a pattern detection task for different detector performance) could be framed – and I bet solved by participants – in a completely social-free framing. Furthermore, the participants are told that there the other players have been recorded, making mental state inference even less useful. This lack of explicit need for ToM in the task (IMHO) also challenges the fMRI "predictions" outlined by the others.

The reviewer raises important points which social neuroscience has grappled with since its inception (e.g., Lockwood et al., *TICS*, 2020). We think there are two issues in play here, which can be decoupled. The first issue is whether theory-of-mind like computations are involved in solving our task. We agree with the reviewer that our task indeed probes how people are inferring a latent state – others' ability – from "overt behaviour". But we disagree that such an inference cannot be considered as requiring theory of mind – indeed, classical theory of mind tasks also require inference on latent states based on overt

behaviour. In fact, we think this is a good characterisation of what classical theory of mind brain networks are engaged in, hence why they are also relevant for our task.

The second issue is whether such inference is uniquely “social” in the sense of requiring inference on another person’s mental states. Here we agree with the reviewer that one could devise an analogous task where the other system is not human but artificial (a pattern detector) – as is true for most social neuroscience studies – and find that the computational problem is solved in a similar manner. However, this result would not change the importance of the identified neurocomputational mechanisms for social behaviour – or affect the conclusions of our manuscript. We do not claim that these neurocomputational mechanisms are uniquely social, although, as we explain below, the neural results strongly indicate that our task is tapping into a set of computations that are foundational to social cognition.

Our winning ToM-model maintains an estimate of the other players’ sensory noise – an unobservable variable which characterises the other players’ internal dynamics and explains variability in their choices. While this variable is different from mental states in the classic theory of mind sense (e.g., false beliefs), our results suggest that it involves similar computations. In particular, the whole-brain other > self contrast identifies the classic theory of mind network (TPJ and dmPFC) – which has been shown to be involved in the representation of others’ mental states – and we find that this network tracks variation in subjects’ confidence in the other players’ choices which is driven by changes in the internal estimate of the other players’ sensory noise (i.e., neural encoding of model-based confidence after controlling for the effect of motion coherence). In addition to the whole-brain contrasts and the model-based analyses, the repetition suppression analysis also shows that TPJ and dmPFC – part of the classic theory of mind network – are differentially activated by other-trials. The reviewer is right that algorithmic short-cuts – which would not be expected to recruit the theory of mind network – might also exist. However, in line with our hypothesis that subjects solve the computational problem in a model-based manner (i.e., using a model of the other players’ internal dynamics), and that this process involves social mental representation, we find that the theory of mind network is indeed recruited in the current study.

Finally, replaying other people’s responses is an approach which we have successfully used before to study social decision-making (e.g., Bang et al., *eLife*, 2020), and we do not believe that this approach nullifies the social nature of our task. Embedding a social task within a cooperative or competitive setting introduces additional factors (e.g., Wittmann et al., *Neuron*, 2016) which may have reduced our ability to identify the core processes involved in a social confidence computation.

In summary, we appreciate the reviewer’s points and agree that these issues often crop up in debates about social neuroscience. We have now revised the manuscript accordingly.

We have made the following changes to the manuscript:

(1.4.A) We now address the issue of social uniqueness/inference in the Discussion [lines 493-505]:

Our study characterises the neurocomputational basis of a common social problem – how we can make predictions about the choice accuracy of another system other than oneself. As is true for most social neuroscience studies, it is possible that one could devise an analogous task where the other system is not human but artificial (e.g., a motion detection algorithm) and find that this task recruits similar neurocomputational mechanisms. However, this result would not alter the importance of these neurocomputational mechanisms for understanding human social behaviour. While our study was not designed to address the issue of social uniqueness⁵⁷, the involvement of a ToM network in the computation of confidence in other players’ choices supports that our task taps into fundamental social processes. In addition, the recruitment of a ToM network supports our interpretation that subjects computed social confidence by estimating the other players’ sensory noise – a process akin to inference about latent mental states which similarly can explain variability in others’ behaviour – and not simply by averaging trial-by-trial feedback about the other players’ choices.

(1.5) Concerning the fMRI results I will make more generic comments on the first results, as other results (e.g., functional connectivity) rely on the validity of the computational model, which I questioned in the points above). My first remark is that the sample size and the trial numbers are both below the standards of the field.

We hope that the new analyses described above have addressed the reviewer's concerns about our modelling approach. Furthermore, we believe that the model recoverability analyses – together with the robust fMRI results – strongly indicate that sample size is not an issue for the current work.

(1.6) The activations of the SELF>Other contrast presents a big confound in that the self condition demands motor preparation and execution. This extra decision and motor execution related to the selection of the chosen option could actually explain most of the neural network revealed in Fig 4 top, which seems to include the preSMA, (wrongly 12labelled as dACC) as well as (left) primary motor cortex (labelled as posterior parietal). This challenges the current authors' interpretations of those activations as "perceptual". Also, I do not get the rationale explaining why classic "perceptual decision-making regions – including MT+ and LIP – [should be] more active on self- than other-trials during the decision phase" (lines 306-307)

We thank the reviewer for the opportunity to clarify these aspects of our results. We use the self versus other whole-brain contrasts to identify brain regions which are differentially activated by self-related versus other-related processing and thereby may play different roles in a non-social versus a social confidence computation. As the reviewer points out, these relatively coarse contrasts are also likely to identify brain regions supporting fundamental aspects of perceptual decision-making – including motor preparation – as subjects are only required to make an active choice on self-trials. In the manuscript, we therefore only use "self-related processing" and "perceptual decision-making" to describe the self > other whole-brain results. We do not interpret a sub-set of the identified activations as "perceptual" until later in the manuscript where we run ROI analyses of MT+ and LIP – identified using a priori functional and anatomical masks. These ROI analyses are motivated by extensive prior work implicating MT+ and LIP in sensory integration. Accordingly, we show that, while MT+ and LIP are overall more activated by self- than other-trials, both ROIs also encode motion coherence on both self- and other-trials. This result – and later analysis of their functional coupling with TPJ/dmPFC – support our interpretation that MT+ and LIP provides the sensory representation that underpins a confidence computation on both trial types. As we hope to have explained above, we are confident that the active choice aspect of the self-trials does not affect this interpretation.

The reviewer raises the interesting point as to why MT+ and LIP were overall more active on self- versus other-trials. The sentence quoted from our manuscript refers simply to the observed whole-brain result and not a prediction or interpretation. In fact, as described in the Introduction, we expected that these regions would be equally active on self- and other-trials – as a sensory representation is required for a confidence computation on both trial types – and were surprised by this difference. However, in line with our prediction, the ROI analyses show that MT+ and LIP do track motion coherence on both self- and other-trials when they are analysed separately. In other words, the difference in baseline activity identified by the whole-brain analysis – which is separate from the neural encoding of motion coherence – is likely to be driven by other factors. One of these factors may indeed be active choice (as we discuss in relation to LIP in response to **point (2.3)**). For example, choice reaction time is thought to be coupled to evidence accumulation, and there may be additional integration of lingering sensory information after termination of the motion stimulus when an active choice is required.

We agree with the reviewer that the illustrative anatomical labels are imprecise. The self > other contrast generated large clusters of activity which spanned multiple brain regions. These clusters included the regions listed above (i.e., dACC and posterior parietal), which we highlighted because of their known involvement in perceptual decision-making, but in hindsight it was imprecise to use these labels for what are in fact much larger clusters. We have therefore removed the illustrative anatomical labels – our ROIs for subsequent analysis are still shown in "**Figure 5: Anatomical masks for ROI analysis**".

We have made the following changes to the manuscript:

(1.6.A) We have refined the description of the whole-brain results and the subsequent ROI predictions to address the difference between active choice (self-trials) and passive viewing (other-trials) and the higher baseline activity in MT+ and LIP on self-trials [lines 277-311]:

To identify brain regions that were differentially activated by self- and other-related processing, we first estimated whole-brain contrasts between self- and other-trials during the decision and the gamble phase (**Figure 1**). During the stimulus phase, the self > other contrast identified classic perceptual decision-making regions, including extrastriate cortex, posterior parietal cortex and cingulate cortex, and – in line with only self-trials requiring active choice – motor regions (**Figure 4**). By contrast, the other > self contrast identified a classic ToM network, including middle temporal gyrus, TPJ and dmPFC (**Figure 4**). During the gamble phase, the self > other contrast did not identify differential activity, whereas the other > self contrast again identified the ToM network (results not shown for gamble phase – see **Table S2** and **Table S3** for all clusters surviving whole-brain correction for decision and gamble phases).

We next focused on a subset of these brain regions, which were defined using independent ROI masks (**Figure 5**), to assess the neural evidence for a ToM account of a social confidence computation. First, we hypothesised that visual motion area MT+, identified in a separate localiser scan, and a human homologue of monkey LIP, encompassing posterior parts of the superior parietal lobule and the intraparietal sulcus³², may support sensory representations of the motion stimulus which informs confidence in both one's own and others' choices. On this account, MT+ and LIP – implicated in sensory integration on the random dot motion task^{8,33–36} – should encode motion coherence on both trial types. We surmise that the higher baseline activity in MT+ and LIP on self-trials (**Figure 4**) was driven by decision processes specific to self-trials such as the requirement for making an active choice. Second, we hypothesised that TPJ and dmPFC – implicated in social inference^{21,22,37} – may combine a sensory representation of the motion stimulus with a distinct representation of others' ability as required by a social confidence computation. On this account, TPJ and dmPFC should track confidence in others' choices after controlling for the sensory representation of motion coherence. We tested these predictions using a complementary analysis approach: we (1) visualised temporally resolved neural encoding profiles by applying sliding-window regressions to up-sampled single-trial ROI activity time courses and then (2) estimated single-trial canonical hemodynamic response functions (c-HRF) for significance testing.

(1.6.B) We have removed the illustrative anatomical labels from “**Figure 4. Whole-brain activations for self- and other-related processing during decision phase**”.

REVIEWER 2

Bang et al studied how confidence in the decision of another person is computed. In a task that manipulated the proficiency of the other “person” at the task and the difficulty of the decision itself, this study shows that observers take both discrimination difficulty and the proficiency of the person being observed into account when computing confidence in others’ choices. Concurrently recorded BOLD signals show that TPJ and dmPFC signals encoded observers’ trial-by-trial estimates of confidence in others and ‘social prediction’ errors, suggesting these areas might be involved in the computation of confidence in others. This is an interesting and timely paper using a novel experimental design. I have a few comments that the authors should be able to address.

(2.1) The models used to fit the behavioral data and explain fMRI responses are all grounded in Bayesian decision theory. However, Bayesian models of confidence have not received strong empirical support as there are many examples of papers showing that confidence ratings do not track the probability of being correct (Koizumi et al., 2015; Maniscalco et al., 2016; Samaha et al., 2016; Zylberberg et al., 2012, 2014) and that Bayesian models do not necessarily fit confidence data the best (Adler & Ma, 2018; Denison et al., 2018; Li & Ma, 2020). To the extent that the model-predictions are used in regression analysis to explain fMRI responses, it would seem that basing the model on incorrect assumptions about how confidence is computed could impact the nature and interpretation of the fMRI results.

We regret that this point was not made more clearly in our manuscript, but the candidate models of confidence in others’ choices included both non-Bayesian (Q) and Bayesian (S, ToM) formulations. In particular, the Q-model forms confidence in others’ choices by tracking the long-run average accuracy achieved by the other player – as opposed to Bayesian inference about the probability that a given choice is correct given the stimulus and their estimated ability.

We used the winning ToM-model for fMRI analysis after demonstrating: (1) that it can predict all features of confidence in others’ choices and (2) that it beats the other candidate models in a quantitative model comparison which controls for model complexity. It is of course possible that there exists another model which provides a (slightly) better model fit. However, our results show that such a model should maintain and combine separate estimates of decision difficulty and others’ ability – a central conclusion of our manuscript. More broadly, our results do not depend on a specific algorithmic implementation of a ToM-like computation, and instead identify the critical qualitative features of this computation.

Finally, to further motivate our approach, we want to highlight that there are many papers which provide empirical support for Bayesian models of confidence (e.g., Aitchison*, Bang* et al., *PloS Computational Biology*, 2015; Fleming & Daw, *Psychological Review*, 2017) – including in the context of the random dot motion task used in our study (e.g., Kiani & Shadlen, *Science*, 2009; Bang & Fleming, *PNAS*, 2018; Fleming et al., *Nature Neuroscience*, 2018; Khalvati et al., *Nature Communications*, 2021). However, we acknowledge that the computational basis of decision confidence is an evolving area of research, and we have now included a discussion of this topic.

We have made the following changes to the manuscript:

(2.1.A) We now address the issue of Bayes optimality in the Discussion [lines 467-478]:

Our confidence models were all grounded in Bayesian decision theory on self-trials, but they differed in their solution to a social confidence computation on other-trials. The Q-model solves this problem in a model-free manner, using running estimates of the other players’ choice accuracy as a proxy for confidence in their choices. In contrast, the best-fitting model, the ToM-model, solves the problem in a Bayesian manner, by integrating distinct representations of decision difficulty and others’ ability. We acknowledge that the computational basis of (non-social) confidence is still debated – with different studies supporting either Bayesian^{7,44–47} or non-Bayesian^{48–50} solutions. However, the grounding of our confidence models in Bayesian decision theory is unlikely to have biased our results. In keeping with standard signal detection theory, we modelled the sensory evidence as a single sample from a Gaussian distribution. Under these constraints, Bayesian and non-Bayesian confidence estimates are monotonically related and will therefore make similar predictions⁶.

(2.2) I found it a bit difficult to understand how specific some effects were as comparison between tasks were not always reported. For instance, for the analysis presented in figure 7 – my understanding is that only ‘other’ trials were used to predict bold responses in TPJ and dmPFC, which showed encoding of model-based estimates of others’ confidence in these regions, but is this response specific to that task or do these regions also show encoding of confidence in general (i.e., in oneself). Since different regions of PFC have been suggested to perform metacognitive functions (Rahnev et al., 2016) it’s unclear if the authors think that dmPFC is specific for evaluating confidence in others. They could report whether the relationship between other’s confidence and TPJ/dmPFC activity is stronger than the relationship between one’s own confidence and TPJ/dmPFC activity.

We thank the reviewer for this suggestion and have now performed this analysis. As shown below, we find that both TPJ and dmPFC encoded an interaction between trial type and model-based confidence estimates: the results suggest that TPJ is selective for social inference, whereas dmPFC supports a separation of self- and other-related information – a result which is in line with recent work on the role of dmPFC in social decision-making (Wittmann et al., *Neuron*, 2016; Wittmann et al., *Neuron*, 2021).

We have made the following changes to the manuscript:

(2.2.A) We have revised the description of the model-based fMRI analysis [lines 351-387]:

Having found that MT+ and LIP track motion coherence on both trial types, we next turned to the social ROIs in order to assess their roles in a social confidence computation. The whole-brain analysis showed that the ToM network – including TPJ and dmPFC – was more active on other- than self-trials during both trial phases. However, a ToM account predicts that this network should not only discriminate between self and other but also differentially encode confidence for self and other. While previous work suggests that TPJ should be selective for a social confidence computation^{39,40}, the potential role of dmPFC is less clear: recent research indicates that dmPFC does not selectively encode social information but instead supports a separation between self- and other-related information when required by the task⁴¹⁻⁴³. To test these hypotheses, we quantified the neural impact of trial type (social; self = -.5 and other = .5), our model-based confidence estimates (as computed under the ToM-model for both self and other) and the interaction between these terms. To control for low-level sensory effects, we orthogonalised the model-based confidence estimates with respect to motion coherence.

This analysis showed that both TPJ and dmPFC encoded an interaction between trial type and the model-based confidence estimates (**Figure 7A**; c-HRF regression; TPJ social: $t(20) = 6.33$, $P < .001$; TPJ confidence, $t(20) = -1.79$, $P = .089$; TPJ interaction, $t(20) = -2.10$, $P = .049$; dmPFC social: $t(20) = 3.70$, $P = .001$; dmPFC confidence, $t(20) = -1.15$, $P = .264$; dmPFC interaction, $t(20) = -3.44$, $P = .003$). However, the form of this interaction effect differed in each of TPJ and dmPFC (**Figure 7B**). In support of a selectivity for social inference, TPJ encoded the model-based confidence estimates on other-trials only, with higher activity when confidence was low (c-HRF regression; confidence on other-trials, $t(20) = -2.17$, $P = .043$; confidence on self-trials, $t(20) = -0.60$, $P = .555$). By contrast, dmPFC encoded model-based confidence estimates on both trial types: activity decreased with confidence on other-trials, but increased with confidence on self-trials, although the effect on self-trials did not reach statistical significance (c-HRF regression; confidence on other-trials, $t(20) = -2.67$, $P = .015$; confidence on self-trials, $t(20) = 1.79$, $P = .089$). The net effect of this inverse response profile in dmPFC is a larger neural distinction between other- and self-trials when confidence is low compared to when confidence is high.

(2.2.B) We have revised “**Figure 7: TPJ and dmPFC support a social confidence computation**” to reflect this new analysis:

Figure 7. TPJ and dmPFC support a social confidence computation. **(A)** The time courses are coefficients from a regression in which we predicted z-scored TPJ (*left*) and dmPFC (*right*) activity time courses using trial type (cyan; other = .5; self = -.5), z-scored model-based confidence estimates as computed under the ToM-model (pink; first orthogonalised with respect to motion coherence) and their interaction (green). The insets show coefficients from an analysis of canonical HRFs (c-HRFs; asterisk indicates statistical significance, $P < .05$, one-sample t-test against zero). **(B)** Same approach as in panel A, but now quantifying the impact of model-based confidence estimates separately for each trial type. **(A-B)** Data are represented as group mean \pm SEM.

(2.2.C) We have also revised the Discussion to reflect these new results [lines 506-536]:

More broadly, our study extends our understanding of the neural basis of social cognition. TPJ and dmPFC have previously been shown to support learning of summary statistics that reflect others' ability on a variety of tasks, including the reliability of advice on a bandit task³⁷, recency-weighted performance on a reaction-time task²¹ or prediction accuracy on a stock-market task²². Our results show that TPJ and dmPFC also support the integration of such knowledge about others' ability with information about the current task state – a process that facilitates accurate predictions about others under conditions of varying task difficulty. There were, however, subtle differences between the activity profiles of TPJ and dmPFC. In line with a hypothesis that TPJ is selective for social inference, TPJ tracked the model-based confidence estimates only on other-trials. By contrast, dmPFC tracked these estimates on both trial types but in an inverse manner – resulting in a larger neural distinction between self and other when confidence was low. This result fits with evidence that dmPFC supports a separation between self- and other-related information in social contexts^{41–43}.

While TPJ involvement was restricted to other-evaluation in the current study, it remains plausible that TPJ may be involved in self-evaluation in other situations. On other-trials, but not self-trials, subjects had to learn a new mapping between the stimulus space and choice accuracy on each block to infer the other's ability. In addition, unlike on self-trials, subjects did not observe the other player's choice and had to consider that either choice could have been made. It remains to be seen whether self-evaluation co-opts putative social brain areas in situations where counterfactual thinking is required⁵⁸ or where one's own ability needs to be learned across time⁵².

(2.3) I am also a bit confused about the interpretation of the LIP area encoding of coherence in self-versus other trials (e.g., figure 6). It's quite striking that LIP encoding of coherence is reversed for self-versus other trials – importantly, this would seem to go against the idea that the same evidence is used for self and other confidence computations. That is, I thought the authors argue that a key variable in the ToM account is a representation of task difficulty akin to computing confidence in one's own choice, but the fact that LIP represents motion evidence so differently in self versus other tasks would seem, at least at first blush, inconsistent with this notion.

The reviewer is correct that LIP encoding of motion coherence is reversed for self- versus other-trials – with negative encoding on self-trials and positive encoding on other-trials. However, as we had hoped to clarify in the main text, this response pattern is consistent with previous work and with our hypothesis that LIP provides a sensory representation for a social confidence computation.

In particular, the LIP response pattern fits with previous observations of posterior parietal fMRI activity during active choice (self) versus passive viewing (other) conditions. When subjects are required to make an active choice, posterior parietal activity decreases with motion coherence (e.g., Hebart et al., *NeuroImage*, 2012). However, during passive viewing, posterior parietal activity increases with motion coherence (Braddick et al., *Perception*, 2001). One explanation of this flip is that LIP neurons integrate sensory information into a belief state over stimulus space (Beck et al., *Neuron*, 2008) – equivalent to the sensory representation in our ToM-model. During active choice, the sensory integration process terminates earlier for high-coherence than low-coherence motion stimuli – a common finding – and bulk neural activity as measured with fMRI is therefore likely to be lower for high-coherence than low-coherence motion stimuli. By contrast, during passive viewing, such early termination of a choice process does not occur and motion coherence itself is likely to drive bulk neural activity. In support of this explanation, we have previously shown that, when motion coherence and choice reaction time are experimentally dissociated during active choice, posterior parietal activity increases with motion coherence, just as on other-trials in the current study (Bang & Fleming, *PNAS*, 2018).

We have now revised the text to clarify that the LIP effects we observe are consistent with (1) previous results from non-social random dot motion tasks and (2) with our hypothesis that LIP provides a sensory representation for a social confidence computation.

We have made the following changes to the manuscript:

(2.3.A) We have revised the description of the LIP results [lines 329-342]:

The LIP response pattern is consistent with previous observations of posterior parietal fMRI activity during active choice (characteristic of self-trials) versus passive viewing (characteristic of other-trials). When subjects are required to make an active choice, posterior parietal activity has been reported to decrease with motion coherence³⁵. In contrast, during passive viewing, posterior parietal activity has been found to increase with motion coherence³⁸. Such a flip is what we would expect if LIP neurons integrate sensory information into a belief state over stimulus space – equivalent to the sensory representation in our computational models. During active choice, the sensory integration process terminates earlier for high-coherence than low-coherence motion stimuli – a common result²³ – and bulk neural activity is therefore likely to be lower for high-coherence than low-coherence motion stimuli. In contrast, during passive viewing, such early termination of a choice process does not occur and motion coherence itself is likely to drive bulk neural activity. In support of this explanation, previous work has shown that, when motion coherence and reaction time are dissociated during active choice, posterior parietal activity increases with motion coherence, just as observed on other-trials⁸.

(2.4) Lastly, was any kind of correction for multiple comparisons used in the time-course analyses? It seems like some differences are present even in the pre-stimulus window (e.g., figure 6)

We did not originally correct for multiple comparisons in the ROI activity time course analyses but agree that treating each time point as independent is too lenient and may return spurious effects such as pre-stimulus encoding of motion coherence. In keeping with standard practice in fMRI analysis, we instead

now perform statistical testing on single-trial ROI activity estimates obtained from fitting a canonical hemodynamic response function to an event of interest separately for each trial. We continue to include ROI activity time courses for temporally resolved and “model-free” visualisation of ROI response profiles. This use of single-trial activity estimates for statistical testing is now well-established in the field (original citation: Atlas et al., *Journal of Neuroscience*, 2010). This new complementary analysis approach (1) allows us to directly address the issue of multiple comparisons over multiple time points while (2) still allowing the reader to inspect the time series associated with reported effects. We highlight that our results are unchanged and thank the reviewer for prompting this revision.

We have made the following changes to the manuscript:

(2.4.A) We have updated the ROI figures to reflect the use of single-trial ROI activity estimates under the canonical hemodynamic response function for significance testing. In addition, we have updated **Figure 6** to match the new analysis of model-based confidence estimates in **Figure 7** prompted by the reviewer in point (2.2). We note that **Figure 8**, which quantified functional connectivity between LIP and TPJ/dmPFC, already established the statistical significance of this coupling independently of the single-trial ROI activity time courses. The updated ROI figures are shown below:

Figure 6. Encoding of motion coherence in MT+ and LIP. **(A)** The time courses are coefficients from a regression in which we predicted z-scored MT+ (*left*) and LIP (*right*) activity time courses using trial type (cyan; other = .5; self = -.5), z-scored motion coherence (pink) and their interaction (green). The insets show coefficients from an analysis of canonical HRFs (asterisk indicates statistical significance, $P < .05$, one-sample t-test against zero). **(B)** Same approach as in panel A, but now quantifying the impact of motion coherence separately for each trial type. **(A-B)** Data are represented as group mean \pm SEM.

Figure 7. TPJ and dmPFC support a social confidence computation. **(A)** The time courses are coefficients from a regression in which we predicted z-scored TPJ (*left*) and dmPFC (*right*) activity time courses using trial type (cyan; other = .5; self = -.5), z-scored model-based confidence estimates as computed under the ToM-model (pink; first orthogonalised with respect to motion coherence) and their interaction (green). The insets show coefficients from an analysis of canonical HRFs (c-HRFs; asterisk indicates statistical significance, $P < .05$, one-sample t-test against zero). **(B)** Same approach as in panel A, but now quantifying the impact of model-based confidence estimates separately for each trial type. **(A-B)** Data are represented as group mean \pm SEM.

Figure 8. Functional coupling between sensory and social ROIs during social confidence computation. **(A)** Schematic of PPI analysis testing whether the correlation between LIP activity (seed region) and TPJ/dmPFC activity is higher on other- than self-trials. **(B)** Contrast estimates from PPI analysis (other > self) as implemented by the Generalised PPI toolbox. **(C)** Visualisation of activity timecourses driving effects documented in panel B. The time courses are coefficients from a regression in which we predicted z-scored TPJ/dmPFC activity time courses using an interaction between z-scored LIP activity time courses and trial type (self = -.5; other = .5), while controlling for the main effect of each term. **(B-C)** Data are represented as group mean \pm SEM.

Figure 9. Encoding of social prediction errors in TPJ and dmPFC. The time courses are coefficients from a regression in which we predicted z-scored TPJ (*left*) and dmPFC (*right*) activity time courses using z-scored social prediction errors as computed under the best-fitting ToM-model. The insets show coefficients from an analysis of canonical HRFs (c-HRFs; asterisk indicates statistical significance, $P < .05$, one-sample t -test against zero). Data are represented as group mean \pm SEM.

(2.4.B) We have revised the reporting of the results accordingly:

[...] We tested these predictions using a complementary analysis approach: we (1) visualised temporally resolved neural encoding profiles by applying sliding-window regressions to up-sampled single-trial ROI activity time courses and then (2) estimated single-trial canonical hemodynamic response functions (c-HRF) for significance testing. [Lines 307-311]

[...] A ToM account predicts that regions supporting the formation of a sensory representation of the motion stimulus should also contribute to a social confidence computation. In other words, we would expect our sensory ROIs to carry information about motion coherence on both self- and other-trials. To test this hypothesis, we quantified the neural impact of trial type (social; now coded as self = -.5 and other = .5), motion coherence and their interaction. As expected under a ToM account, MT+ and LIP tracked motion coherence on both trial types (**Figure 6**). In MT+, the response profile was the same on self- and other-trials: the higher the coherence, the higher the activity (c-HRF regression; social: $t(20) = -7.72$, $P < .001$; coherence, $t(20) = 5.42$, $P < .001$; interaction, $t(20) = -0.85$, $P = .408$; coherence on other-trials, $t(20) = 3.94$, $P < .001$; coherence on self-trials, $t(20) = 3.65$, $P = .002$). In contrast, the LIP response profile differed between trial types: the higher the coherence, the higher the activity on other-trials, but the lower the activity on self-trials (c-HRF regression; social: $t(20) = -7.86$, $P < .001$; coherence, $t(20) = 0.20$, $P = .844$; interaction, $t(20) = 3.07$, $P = .006$; coherence on other-trials, $t(20) = 2.94$, $P = .008$; coherence on self-trials, $t(20) = -2.14$, $P = .045$). [Lines 315-328]

[...] This analysis showed that both TPJ and dmPFC encoded an interaction between trial type and the model-based confidence estimates (**Figure 7A**; c-HRF regression; TPJ social: $t(20) = 6.33$, $P < .001$; TPJ confidence, $t(20) = -1.79$, $P = .089$; TPJ interaction, $t(20) = -2.10$, $P = .049$; dmPFC social: $t(20) = 3.70$, $P = .001$; dmPFC confidence, $t(20) = -1.15$, $P = .264$; dmPFC interaction, $t(20) = -3.44$, $P = .003$). However, the form of this interaction effect differed in each of TPJ and dmPFC (**Figure 7B**). In support of a selectivity for social inference, TPJ encoded the model-based confidence estimates on other-trials only, with higher activity when confidence was low (c-HRF regression; confidence on other-trials, $t(20) = -2.17$, $P = .043$; confidence on self-trials, $t(20) = -0.60$, $P = .555$). By contrast, dmPFC encoded model-based confidence estimates on both trial types: activity decreased with confidence on other-trials, but increased with confidence on self-trials, although the effect on self-trials did not reach statistical significance (c-HRF regression; confidence on other-trials, $t(20) = -2.67$, $P = .015$; confidence on self-trials, $t(20) = 1.79$, $P = .089$). The net effect of this inverse response profile in dmPFC is a larger neural distinction between other- and self-trials when confidence is low compared to when confidence is high. [Lines 373-387]

[...] Finally, having examined how sensory and social regions interact to compute confidence in others' choices, we analysed how the representation of another player's ability is updated with task experience.

The behavioural and modelling results indicated that subjects revised their estimate of the other player's sensory noise based on the difference between the accuracy of the other player's choice and their confidence in this choice being correct (see schematic of learning mechanism in **Figure 3**). Consistent with a role in supporting this estimate, both TPJ and dmPFC tracked social prediction errors as computed under the ToM-model (**Figure 9**; (c-HRF regression; TPJ, $t(20) = -3.18$, $P = .005$; dmPFC, $t(20) = -3.12$, $P = .005$). We note that these response profiles remained after orthogonalising the social prediction error with respect to either the outcome term (accuracy) or the prediction term (confidence). [Lines 409-418]

(2.4.C) We have revised the description in the Methods of how the single-trial ROI activity estimates under the canonical hemodynamic response function were generated (note that we already used this approach for the repetition suppression analysis in **Figure S7**) [lines 860-874]:

In addition to the whole-brain GLM analyses described above, we also estimated whole-brain GLMs which included a separate regressor for each trial⁷⁰. For the stimulus-related analyses (**Figure 6**, **Figure 7** and **Figure S7**), the regressors were boxcars spanning RDK presentation. For the feedback-related analysis (**Figure 9**), the regressors were boxcars spanning feedback delivery. Each of the regressors was convolved with a canonical hemodynamic response function. We included motion parameters as 'nuisance' regressors. Regressors were modelled separately for each scan run and constants were included to account for between-run differences in mean activation and scanner drifts. A high-pass filter (128 s cut-off) was applied to remove low-frequency drifts. One consideration when obtaining single-trial activity estimates as a beta time series is that a beta for a given trial can be affected by acquisition artefacts that occur together with that trial (e.g., scanner pulse artefacts). Therefore, for each subject, we computed the grand-mean beta estimate across both voxels and trials and excluded any trial whose mean beta estimate across voxels was 3 SDs below or above this grand mean⁷⁰. Finally, we used the ROI masks to extract single-trial ROI activity estimates under the canonical hemodynamic response function.

REVIEWERS' COMMENTS

Reviewer #1 (Remarks to the Author):

I am glad to see that the authors have addressed all my concerns. I believe the paper is in much better shape now. I have no remaining concerns.

Reviewer #3 (Remarks to the Author):

The authors have adequately addressed my prior comments with substantial revisions and re-analyses.

Reviewer #1 (Remarks to the Author):

I am glad to see that the authors have addressed all my concerns. I believe the paper is in much better shape now. I have no remaining concerns.

We thank the reviewer for the positive assessment and their constructive comments in the previous round of review.

Reviewer #3 (Remarks to the Author):

The authors have adequately addressed my prior comments with substantial revisions and re-analyses.

We thank the reviewer for the positive assessment and their constructive comments in the previous round of review.